# Host-microbiome metabolism of a plant toxin in bees

Erick VS Motta[1]*, Alejandra Gage[1], Thomas E Smith[1], Kristin J Blake[2], Waldan K Kwong[3], Ian M Riddington[2], Nancy Moran[1]*

[1]Department of Integrative Biology, The University of Texas at Austin, Austin, United States; [2]Mass Spectrometry Facility, Department of Chemistry, The University of Texas at Austin, Austin, United States; [3]Instituto Gulbenkian de Ciência, Oeiras, Portugal

**Abstract** While foraging for nectar and pollen, bees are exposed to a myriad of xenobiotics, including plant metabolites, which may exert a wide range of effects on their health. Although the bee genome encodes enzymes that help in the metabolism of xenobiotics, it has lower detoxification gene diversity than the genomes of other insects. Therefore, bees may rely on other components that shape their physiology, such as the microbiota, to degrade potentially toxic molecules. In this study, we show that amygdalin, a cyanogenic glycoside found in honey bee-pollinated almond trees, can be metabolized by both bees and members of the gut microbiota. In microbiota-deprived bees, amygdalin is degraded into prunasin, leading to prunasin accumulation in the midgut and hindgut. In microbiota-colonized bees, on the other hand, amygdalin is degraded even further, and prunasin does not accumulate in the gut, suggesting that the microbiota contribute to the full degradation of amygdalin into hydrogen cyanide. In vitro experiments demonstrated that amygdalin degradation by bee gut bacteria is strain-specific and not characteristic of a particular genus or species. We found strains of *Bifidobacterium*, *Bombilactobacillus,* and *Gilliamella* that can degrade amygdalin. The degradation mechanism appears to vary since only some strains produce prunasin as an intermediate. Finally, we investigated the basis of degradation in *Bifidobacterium* wkB204, a strain that fully degrades amygdalin. We found overexpression and secretion of several carbohydrate-degrading enzymes, including one in glycoside hydrolase family 3 (GH3). We expressed this GH3 in *Escherichia coli* and detected prunasin as a byproduct when cell lysates were cultured with amygdalin, supporting its contribution to amygdalin degradation. These findings demonstrate that both host and microbiota can act together to metabolize dietary plant metabolites.

*For correspondence:
erickvsm@utexas.edu (EVSM);
nancy.moran@austin.utexas.edu
(NM)

Competing interest: The authors declare that no competing interests exist.

## Editor's evaluation

The manuscript makes an important contribution to understanding the roles of the bee host and microbiome in degrading amygdalin, a dietary secondary metabolite. Several bacterial strains and their enzymes responsible for the deglycosylation of amygdalin are identified. Conclusions are reached convincingly through a comprehensive combination of in vitro and in vivo experiments including gene-expression analysis, proteomics, HPLC-MS, and the use of recombinant *E. coli* to test enzyme function. The consequences of microbial-derived amygdalin metabolisation on host health remain uncertain from the experiments conducted, but this work should stimulate future research into the importance of secondary metabolite processing by the microbiome on insect host health.

## Introduction

Many animals ingest potential toxins along with their food, and these toxins can have complex consequences. Dietary toxins are often deleterious, but they sometimes prove beneficial, by providing protection against natural enemies, including pathogens and parasites (*Gowler et al., 2015*). Once

**eLife digest** Most plants produce chemicals that are toxic to at least some animals. Whether or not the toxins are harmful to a particular animal depends on how much they consume and the specific biochemistry that occurs during digestion. The enzymes produced in the gut both by the animal and by the microbes that reside there often help break down toxic substances into less harmful molecules. However, some products of this breakdown can be toxic themselves. While these products can harm the animal, they may also be detrimental to parasites living in the gut, resulting in an overall positive effect.

Almonds and their pollen are consumed by humans and bees without apparent harmful effects. However, almonds contain amygdalin, a molecule that can produce the highly toxic compound hydrogen cyanide upon digestion. Although amygdalin can be toxic to bees in high doses, the amount usually found in almond nectar is not harmful, and indeed, it may protect bees from parasites. Motta et al. wanted to know how amygdalin is digested in the gut of bees, and whether gut microbes have a role in this digestion.

To answer these questions, Motta et al. compared the effects of consuming amygdalin on normal bees and bees lacking gut microbes. Bees without gut microbes broke down amygdalin into a harmless substance called prunasin. However, only bees with gut microbes could further break down prunasin into hydrogen cyanide. Interestingly, the full metabolism of amygdalin had no detectable effect on whether the bees survived for longer times or on which microbes were found in the gut. Motta et al. also found some gut bacteria in bees that can break down amygdalin and release hydrogen cyanide, and identified the enzyme responsible for the process. When the gene encoding this enzyme was inserted into a different species of bacteria, the second species gained the ability to break down amygdalin.

The findings of Motta et al. explain a role of gut microbes in processing amygdalin in bees. In the future, this may be the key to understanding how humans and other creatures process plant toxins. Future work on the relationship between animals and microbes living in their guts could help scientists understand how to manipulate the digestion and processing of toxins, nutrients, or drugs to benefit human health.

---

ingested, enzymatic metabolism of dietary compounds can render them more or less toxic (*Mason et al., 2019*; *Dearing and Weinstein, 2022*). The host itself can produce enzymes that degrade toxins. Additionally, gut microbiota members have been shown to contribute to enzymatic degradation of dietary compounds, including toxins.

As generalist foragers, honey bees and bumble bees can be exposed to a wide range of plant secondary metabolites (*Irwin et al., 2014*), which are usually produced by plants as defenses against pathogens and herbivores (*Zaynab et al., 2018*). Even when in low concentrations, these metabolites can have a range of effects on bee behavior and health, from negative to neutral to positive, and can be involved in attraction or deterrence (*Detzel and Wink, 1993*; *Hagler and Buchmann, 1993*; *Stephenson, 1982*). Interestingly, some bee species cannot detect naturally occurring concentrations of certain nectar metabolites, such as quinine, nicotine, caffeine, and amygdalin (*Tiedeken et al., 2014*). This poor acuity may lead to long-term side effects depending on the toxicity of the metabolite.

A plant secondary metabolite to which generalist bees may be chronically exposed is amygdalin, a cyanogenic glycoside found in almonds, apples, cherries, and nectarines (*London-Shafir et al., 2003*; *Barceloux, 2009*; *Bolarinwa et al., 2015*; *Lee et al., 2017*). Studies on almonds show that amygdalin is present in nectar and pollen (*London-Shafir et al., 2003*). The western honey bee, *Apis mellifera*, is the primary pollinator of almonds and likely also encounters amygdalin in other crops. The toxicity of amygdalin derives from its degradation products (*Jaszczak-Wilke et al., 2021*). Degradation occurs during plant tissue damage, such as chewing by herbivores, since amygdalin is stored in cell vacuoles and the glycoside hydrolases (GHs) involved in degradation are present in the cytoplasm. During degradation, amygdalin is usually first broken down into prunasin and a glucose molecule. Then, prunasin is broken down into another glucose molecule and mandelonitrile, with the latter compound converted into benzaldehyde and hydrogen cyanide. Hydrogen cyanide is a toxic molecule that can lead to acute poisoning in animals (*Khandekar and Edelman, 1979*; *Carter*

*et al., 1980*; *Newton et al., 1981*; *Kolesarova et al., 2021*; *Kovacikova et al., 2019*; *Salama et al., 2019*) as it interferes with the electron transport chain during oxidative phosphorylation (*Cooper and Brown, 2008*).

Interestingly, some bees are not deterred by amygdalin concentrations encountered in almond nectar (up to 15 µM) (*London-Shafir et al., 2003*; *Stevenson et al., 2017*), and can tolerate concentrations up to 219 µM with no effects on survivorship (*Irwin et al., 2014*; *Lecocq et al., 2018*). Bees feeding on particular plants may be exposed to even higher doses of amygdalin; for example, concentrations in almond pollen can reach up to 4 mM (*London-Shafir et al., 2003*). Exposure to these high doses of amygdalin leads to acute malaise symptoms, including a sharp increase in time spent upside down and abdomen dragging (*Hurst et al., 2014*), and exposure for several days lowers bee survivorship (*Kevan and Ebert, 2005*). Lower doses can also result in lower survivorship in lab trials (*Ayestaran et al., 2010*). Despite the potential for toxicity to bees, colony-level exposure to amygdalin may protect bees against parasites, such as the trypanosomatid *Lotmaria passim* (*Tauber et al., 2020*), and may reduce the titer of some pathogenic viruses (*Tauber et al., 2020*; *Palmer-Young et al., 2017*). Thus, amygdalin exposure may have both positive and negative effects on bee health depending on dose and infection status.

Despite potential consequences for bee health, the routes of amygdalin metabolism within bees have not been elucidated. The genomes of honey bees and bumble bees have fewer detoxification genes compared to other insects (*Berenbaum and Johnson, 2015*; *Sadd et al., 2015*), but they do encode some enzymes that can degrade plant metabolites, such as cytochrome P450 monooxygenases, glutathione transferases, and GHs (*Berenbaum and Johnson, 2015*; *Pontoh and Low, 2002*; *Rand et al., 2015*). For example, honey bees secrete a GH into their mouths from their hypopharyngeal glands that is then transferred to the midgut where it can potentially catalyze the initial breakdown of glycosides (*Pontoh and Low, 2002*; *Ricigliano et al., 2017*), such as the conversion of amygdalin into prunasin.

Amygdalin toxicity occurs after ingestion, but not after injection into the hemolymph (*Hurst et al., 2014*), suggesting that enzymes in the gut achieve the conversion of amygdalin into hydrogen cyanide. The source of these enzymes is unknown. Possibilities include bee GHs (*Pontoh and Low, 2002*), pollen-derived GHs that bees ingest (*Ricigliano et al., 2017*), or GHs produced by the bee gut microbiota (*Kwong and Moran, 2016*; *Zheng et al., 2018*; *Motta et al., 2022a*). The latter possibility is suggested by the vast arsenal of GHs produced by the dominant bee gut bacterial species (*Zheng et al., 2019*; *Ellegaard et al., 2019*). Interestingly, amygdalin itself does not show antibacterial effects in vitro, and the honey bee gut microbiota appears not to be significantly affected by amygdalin exposure (*Tauber et al., 2020*).

In this study, we investigated the contributions of honey bees and their microbiota to amygdalin degradation. We found that breakdown to prunasin is achieved by hosts without a microbiota and that further degradation can be performed by specific strains of dominant microbiota species. Using biochemical assays, we characterized a GH secreted by bee-associated *Bifidobacterium* strains that can degrade amygdalin and prunasin. These findings shed light on how the combined contributions of host and microbiome enable degradation of a dietary plant metabolite.

## Results

To investigate amygdalin metabolism by bee gut bacteria, we selected representative strains of four bacterial groups involved in food metabolism in the bee gut: *Bifidobacterium*, *Bombilactobacillus* (formerly called *Lactobacillus* Firm-4), *Lactobacillus* nr. *melliventris* (formerly called *Lactobacillus* Firm-5), and *Gilliamella* (*Figure 1*). We cultured these strains in semi-defined media (SDM, *Figure 1A*) or in nutritionally rich media (MRS or Insectagro, *Figure 1B*) to assess their susceptibility to amygdalin and their ability to metabolize amygdalin into byproducts, such as prunasin, as analyzed by LC-MS (*Figure 1C*).

### Bee gut bacterial symbionts vary in susceptibility to amygdalin

Within species (or closely related species clusters), strains varied in their ability to cope with different concentrations of amygdalin in vitro.

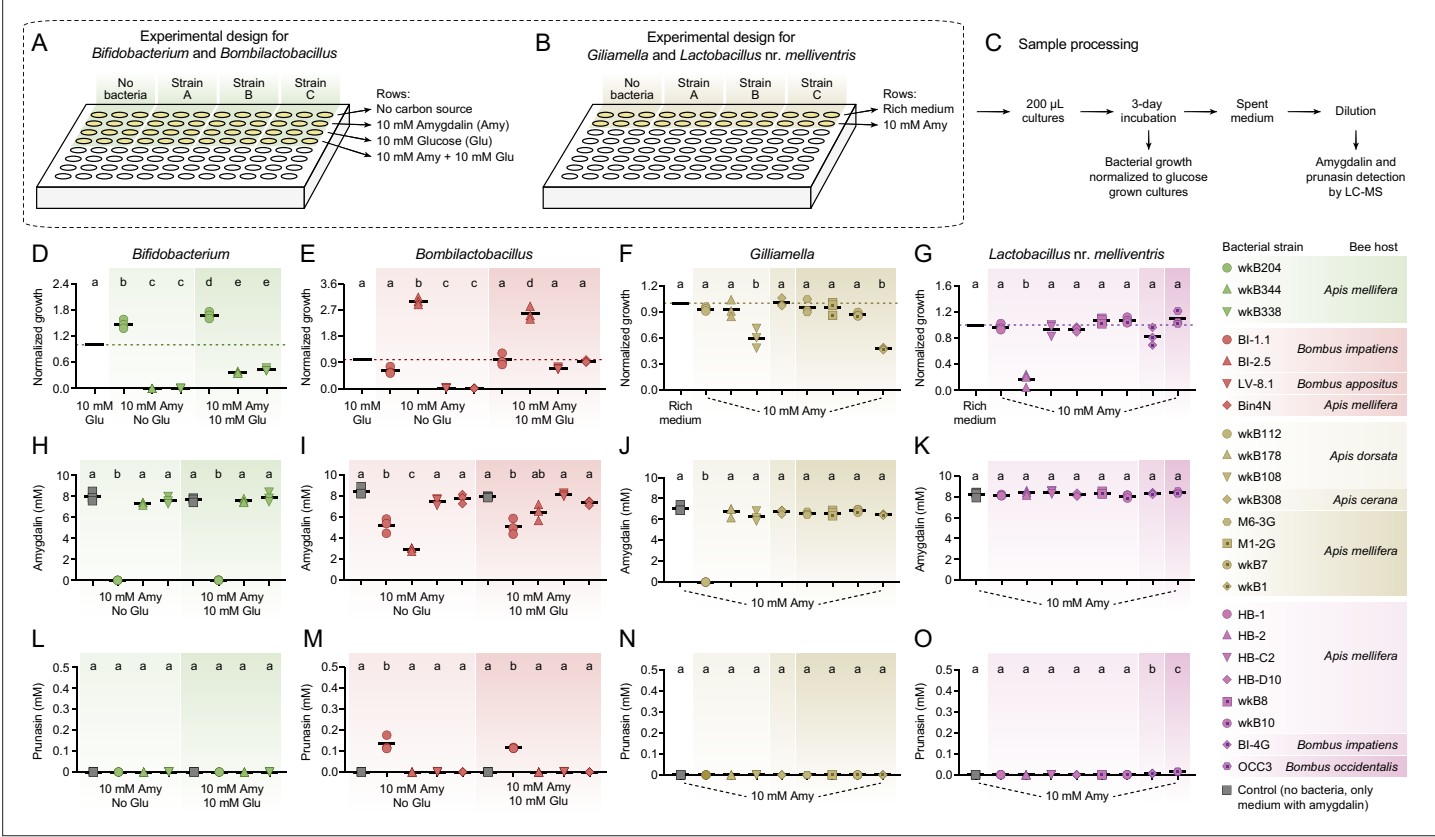

**Figure 1.** In vitro exposure of bee gut bacteria to amygdalin. Experimental design in (**A**) semi-defined or (**B**) nutritionally rich media in 96-well plates. (**C**) Sample processing for LC-MS analysis. (**D**) *Bifidobacterium* and (**E**) *Bombilactobacillus* growth in semi-defined media in the presence of amygdalin (or amygdalin and glucose) normalized to growth in the presence of glucose. (**F**) *Gilliamella* and (**G**) *Lactobacillus* nr. *melliventris* growth in nutritionally rich media in the presence of amygdalin normalized to growth in the absence of amygdalin. Bacterial growth was measured as optical density at 600 nm after 3 days of incubation at 35°C and 5% $CO_2$. (**H–K**) Amygdalin and (**L–O**) prunasin concentrations in spent medium of amygdalin (or amygdalin and glucose) grown cultures of *Bifidobacterium*, *Bombilactobacillus*, *Gilliamella,* and *Lactobacillus* nr. *melliventris*, respectively. Controls consisted of media with amygdalin (or amygdalin and glucose) but no bacteria. Experiments were performed in three biological replicates. Groups with different letters are significantly different (p < 0.01, one-way ANOVA test followed by Tukey's multiple-comparison test).

The online version of this article includes the following figure supplement(s) for figure 1:

**Figure supplement 1.** In vitro exposure of bee gut associated bacteria to 100 mM amygdalin.

## Bifidobacterium

Three strains isolated from the guts of *A. mellifera* (wkB204, wkB344, and wkB338) were cultured in the presence of amygdalin and/or glucose as sole carbon sources in SDM (**Figure 1A**). Strain wkB204 grew in the presence of amygdalin as the sole carbon source, suggesting that this strain degrades amygdalin and is not susceptible to the potential byproducts (**Figure 1D**). On the other hand, strains wkB344 and wkB338 grew only when glucose was added, and their growth was hampered if amygdalin was added, indicating a toxic effect on these strains (**Figure 1D**).

## Bombilactobacillus

Strains isolated from the guts of *Bombus impatiens* (BI-1.1 and BI-2.5), *Bombus appositus* (LV-8.1) and *A. mellifera* (Bin4N) were tested in SDM (**Figure 1A**). Strains BI-1.1 and BI-2.5 grew in the presence of amygdalin as the sole carbon source, with BI-2.5 growing better than BI-1.1. In fact, growth of BI-2.5 was higher with amygdalin than with glucose as the sole carbon source (**Figure 1E**). Strains LV-8.1 and Bin4N grew only in the medium with glucose, but their growth was not affected when amygdalin was added (**Figure 1E**).

### Gilliamella

Strains isolated from the guts of *Apis dorsata* (wkB112, wkB178, and wkB108), *Apis cerana* (wkB308), and *A. mellifera* (M6-3G, M1-2G, wkB7, and wkB1) were cultivated in Insectagro due to the lack of an SDM for these strains (*Figure 1B*). Most of the strains grew at similar rates in the presence or absence of 10 mM amygdalin, except for wkB108 and wkB1 which exhibited a delay in growth, suggesting susceptibility to amygdalin at the tested concentration (*Figure 1F*).

### Lactobacillus nr. melliventris

Strains isolated from the guts of *A. mellifera* (HB-1, HB-2, HB-C2, HB-D10, wkB8, and wkB10), *B. impatiens* (BI-4G), and *Bombus occidentalis* (OCC3) were cultivated in rich medium (MRS) since they do not grow well in SDM (*Figure 1B*). All strains grew in the presence of amygdalin, though HB-2 growth was reduced by adding amygdalin to MRS (*Figure 1G*).

For most bacterial strains tested, growth was hampered by increasing the concentration of amygdalin from 10 to 100 mM (*Figure 1—figure supplement 1A–C*). This toxicity is probably related to the presence of amygdalin itself and not to potential byproducts since most strains could not degrade amygdalin. The amygdalin concentrations were chosen to correspond to the glucose concentrations usually added to growth media to investigate carbon source usage by bacteria, and are higher than the concentrations detected in almond pollen (~4 mM) and nectar (~0.01 mM) (*London-Shafir et al., 2003*).

## Specific bee gut bacterial strains degrade amygdalin

Using LC-MS analyses, amygdalin degradation was confirmed for strains that could grow in the presence of amygdalin as the sole carbon source, such as *Bifidobacterium* strain wkB204 (*Figure 1H*) and *Bombilactobacillus* strains BI-1.1 and BI-2.5 (*Figure 1I*). Amygdalin was not detected (*Figure 1H*) or was detected in a lower concentration (*Figure 1I* and *Figure 1—figure supplement 1D–E*) in the spent medium of amygdalin-grown cultures when compared to the initial concentration. In these cases, amygdalin degradation was observed regardless of whether glucose was present. Interestingly, *Bombilactobacillus* strain BI-2.5 degrades less amygdalin when glucose is also present in the medium (*Figure 1I*). For *Bifidobacterium* strain wkB204 and *Bombilactobacillus* strain BI-1.1, on the other hand, similar levels of amygdalin degradation were detected in cultures with or without glucose (*Figure 1H–I*).

*Gilliamella* and *Lactobacillus* nr. *melliventris* strains were cultivated in nutritionally rich media, and therefore amygdalin degradation was primarily investigated by LC-MS of spent medium. We observed amygdalin degradation only for *Gilliamella* strain wkB112 (*Figure 1J* and *Figure 1—figure supplement 1F*). The use of nutritionally rich media for these strains may have masked the ability of some strains to degrade amygdalin, as they had glucose as an alternative carbon source (*Figure 1J–K*).

## Different mechanisms of amygdalin degradation by bee gut bacteria

Metabolism of amygdalin by *Bifidobacterium* strain wkB204 and *Bombilactobacillus* strain BI-1.1 produces prunasin as a byproduct (*Figure 1L–M* and *Figure 1—figure supplement 1G–H*), although prunasin was only detected in wkB204 cultures after providing an excessive amount of amygdalin (*Figure 1—figure supplement 1G*). This suggests that wkB204 and BI-1.1 encode enzymes to break down the glycosidic bond between the glucose residues in the amygdalin structure, releasing prunasin and one glucose molecule, which can then be used as carbon source by these bacteria. On the other hand, prunasin was not produced by *Bombilactobacillus* strain BI-2.5 or *Gilliamella* strain wkB112 (*Figure 1M–N*) even after adding excess amygdalin (*Figure 1—figure supplement 1H–I*). Therefore, BI-2.5 and wkB112 seem to metabolize amygdalin in a different way than wkB204 and BI-1.1, probably by breaking down the glycosidic bond that links the two glucose residues to the aglycone, releasing a disaccharide and mandelonitrile into the medium. These mechanisms are corroborated by LC-MS analyses of spent medium taken from these cultures on a daily census (*Figure 2*). These results suggest that amygdalin breakdown via a prunasin intermediate is limited to wkB204 and BI-1.1.

## Characterizing an enzyme involved in amygdalin metabolism

After finding that specific strains from different bee gut bacterial species can degrade amygdalin, we focused on honey bee-associated *Bifidobacterium* strains to investigate the enzyme involved in

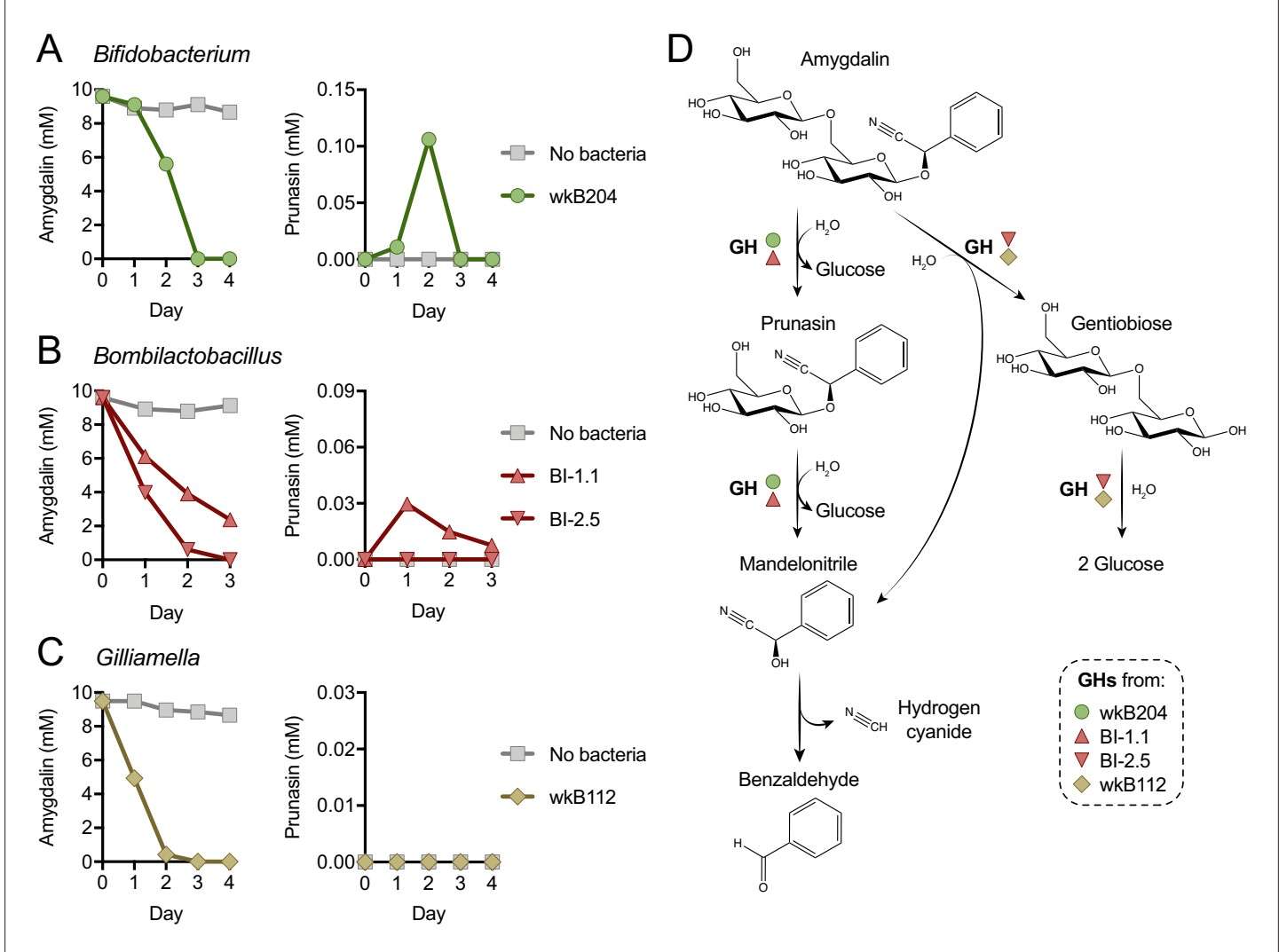

**Figure 2.** Mechanism of amygdalin degradation by bee gut bacteria. Amygdalin and prunasin concentrations detected by LC-MS in spent-medium of 3- or 4-day-old cultures of (**A**) *Bifidobacterium* strain wkB204, (**B**) *Bombilactobacillus* strains BI-1.1 and BI-2.5, and (**C**) *Gilliamella* strain wkB112. Concentrations were determined every day for 3–4 days. Controls consisted of medium with amygdalin but no bacteria. Only wkB204 and BI-1.1 produced prunasin as an intermediate. (**D**) Proposed mechanism of amygdalin degradation by different bacterial species in the bee gut.

this metabolism. First, we checked whether the enzyme is secreted or not. Large cultures of wkB204, wkB344, and wkB338 were grown for 5 days (*Figure 3A*), after which we performed biochemical assays with both spent medium and cell lysate of glucose- and amygdalin-grown cultures.

As observed in the previous experiment, wkB204 completely degraded amygdalin; we did not detect amygdalin in spent medium (10A sm, *Figure 3B*) or in cell lysate of amygdalin-grown cultures (10A cl, *Figure 3C*). To investigate whether the enzyme involved in amygdalin degradation was secreted, we added fresh amygdalin to sterile spent medium (10A sm + 10A) or to sterile cell lysate (10A cl + 10A) originating from amygdalin-grown cultures. After 3 days of incubation, we found full degradation of amygdalin in spent medium (10A sm + 10 A) (*Figure 3B*), but only slight degradation in cell lysate (10A cl + 10A) (*Figure 3C*); this was compared to a control sample containing only medium and amygdalin (Fresh 10A). No amygdalin was detected in cell lysates of amygdalin-grown cultures, showing that amygdalin does not enter bacterial cells (10A cl, *Figure 3C*). Moreover, we detected prunasin in both spent medium and cell lysate of amygdalin-grown cultures supplemented with amygdalin (10A sm + 10A and 10A cl + 10A, respectively) (*Figure 3—figure supplement 1*).

For comparison, these assays were also performed for wkB344 and wkB338 cultures. Some amygdalin degradation was observed for the spent medium of wkB344 amygdalin-grown cultures, but

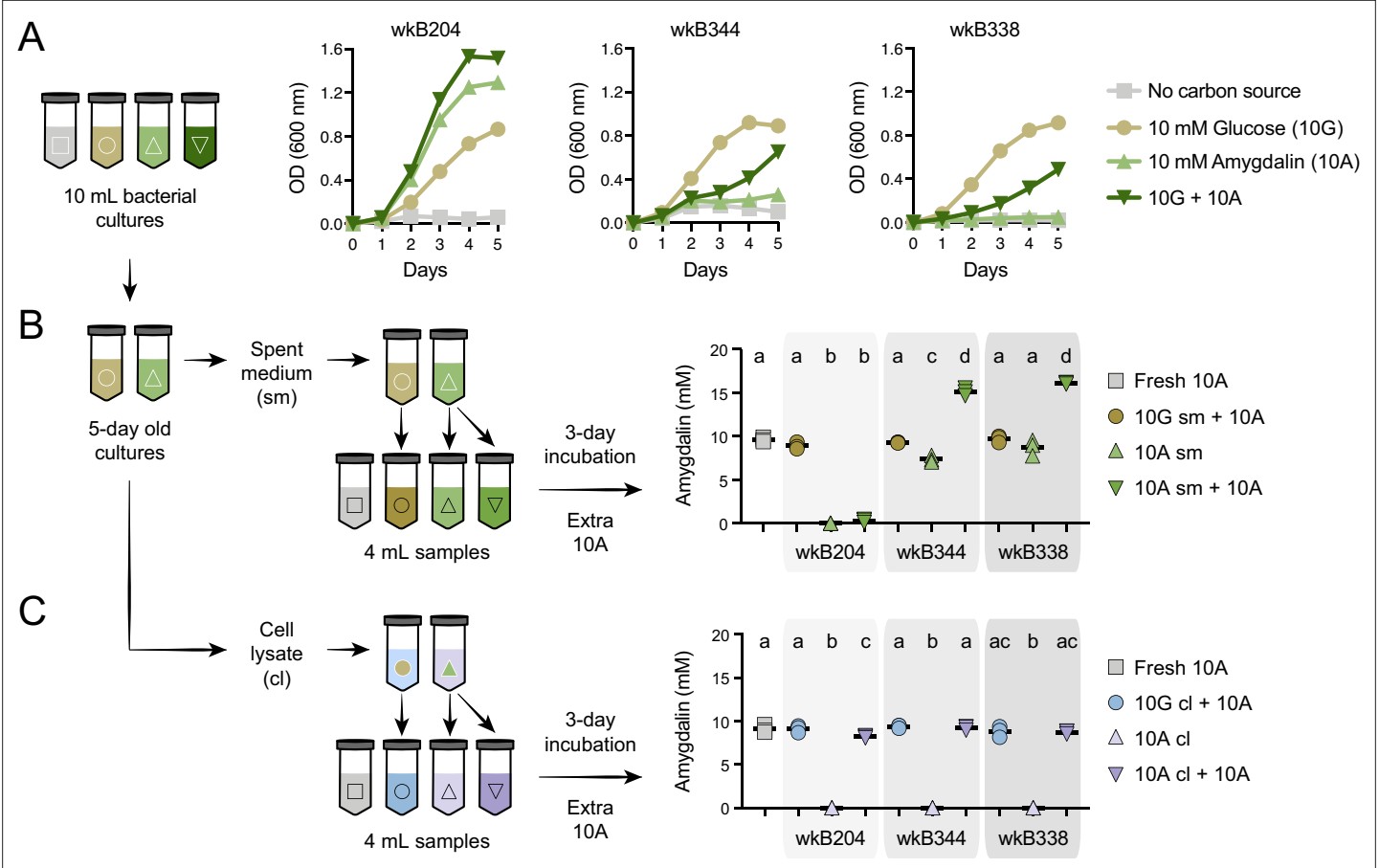

**Figure 3.** Amygdalin degradation in spent media and cell lysates of *Bifidobacterium* strains. (**A**) Bacterial growth curves of *Bifidobacterium* strains cultured in semi-defined media (SDM) without a carbon source, with 10 mM glucose (10G), with 10 mM amygdalin (10A), or with both 10 mM glucose and 10 mM amygdalin (10G+10A) as carbon sources at 35°C and 5% $CO_2$. Experiments were performed in three biological replicates. Each data point represents the average optical density (600 nm) measured every day for 5 days. (**B–C**) For each strain, 10G and 10A grown cultures were separated into (**B**) spent medium (sm), originating from samples 10G sm and 10A sm, and (**C**) cell lysate (cl), originating from samples 10G cl and 10 A cl. These samples were used to investigate amygdalin degradation by adding extra 10A to the samples. Controls consisted of 10A grown cultures without adding extra 10A and fresh SDM with 10A. Reactions were incubated at 35°C and 5% $CO_2$ for 3 days, after which amygdalin concentration was determined. Experiments were performed in three biological replicates. Groups with different letters are significantly different ($p < 0.01$, one-way ANOVA test followed by Tukey's multiple-comparison test).

The online version of this article includes the following figure supplement(s) for figure 3:

**Figure supplement 1.** Prunasin concentrations in spent media and cell lysates of *Bifidobacterium* strains.

this degradation was much less than that observed for wkB204 (*Figure 3B–C*). We also investigated enzyme production and activity in the absence of amygdalin, by adding amygdalin to spent medium and cell lysate from glucose-grown cultures (10G sm + 10A and 10G cl + 10A, *Figure 3B–C*). None of the strains was able to significantly degrade amygdalin under these conditions.

Since spent medium of wkB204 amygdalin-grown cultures achieved full degradation of amygdalin, we decided to characterize the secreted enzyme involved in the degradation. Spent media from wkB204 cultures, grown with either amygdalin or glucose, were processed to obtain concentrated protein extracts (*Figure 4A*). Protein profiles were first obtained by SDS-PAGE gel and showed that amygdalin-grown cultures had a distinct secretome when compared to glucose-grown cultures (*Figure 4B*, *Figure 4—source data 1*). Then, samples were submitted to proteomics analysis, which confirmed the expression differences, as we found 107 proteins secreted in higher abundance in amygdalin-grown cultures and 131 proteins secreted in higher abundance in glucose-grown cultures ($p<0.05$, t-test followed by Benjamini-Hochberg procedure to control for false discovery, *Figure 4C*). Several significantly upregulated proteins in amygdalin-grown cultures are associated with

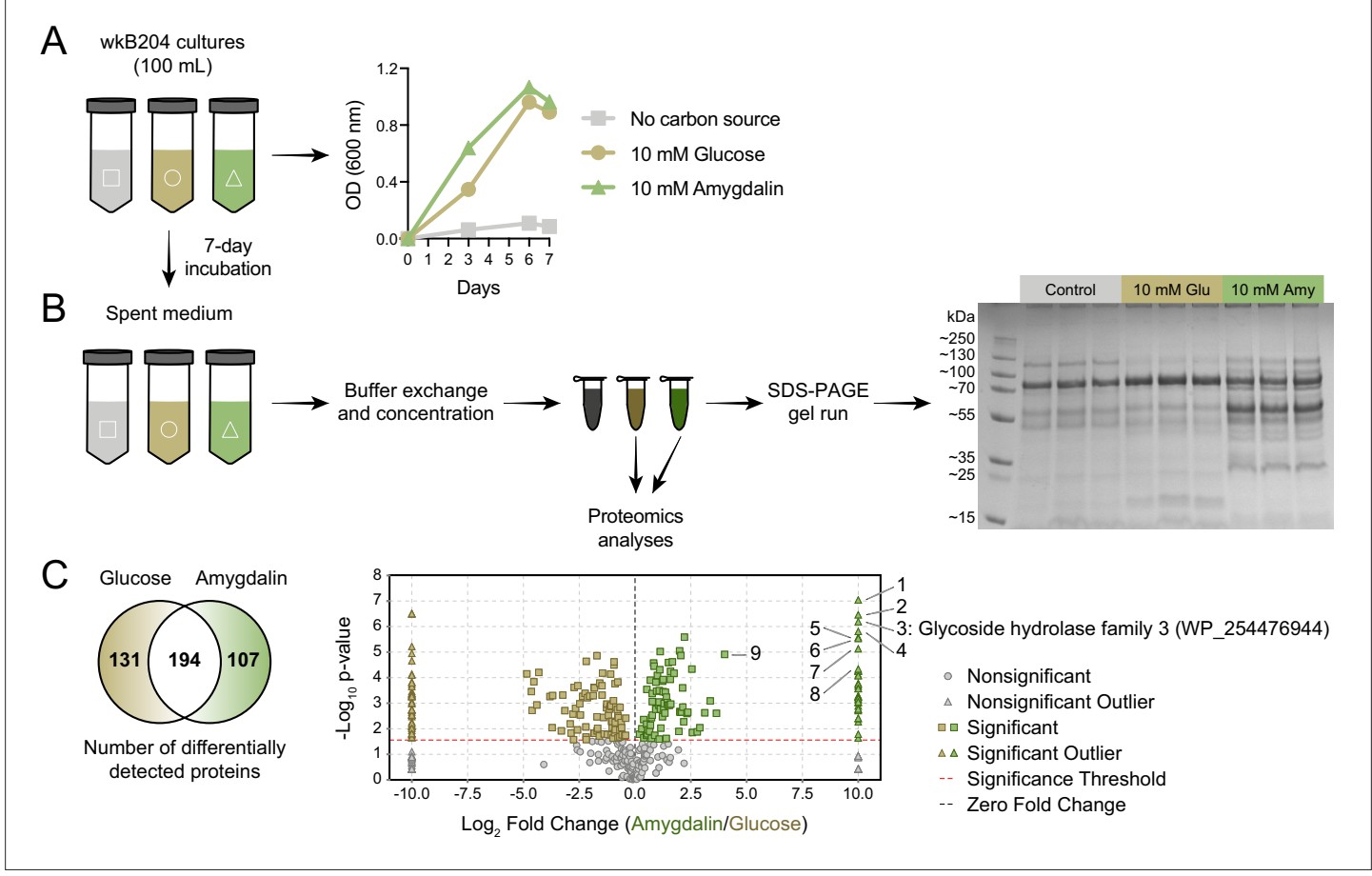

**Figure 4.** Identification of an amygdalin degrading enzyme from *Bifidobacterium*. (**A**) Large-scale culture of *Bifidobacterium* strain wkB204 in semi-defined media (SDM) without a carbon source, with 10 mM glucose, or with 10 mM amygdalin at 35°C and 5% $CO_2$. Experiments were performed in three biological replicates and each data point represents the average optical density (600 nm) measured every day for 7 days. (**B**) Spent medium concentration for running on an SDS-PAGE gel. (**C**) Venn diagram and volcano plot showing the number of differentially expressed proteins in spent medium of glucose- or amygdalin-grown cultures. Numbers in the volcano plot: 1: alpha/beta fold hydrolase (WP_254477374), 2: nucleoside hydrolase (WP_254477231), 3: glycoside hydrolase family 3 (WP_254476944), 4: beta-galactosidase (WP_254477161), 5: alpha-mannosidase (WP_254477012), 6: Nudix hydrolase (WP_254477413), 7: MFS transporter (WP_254476943), 8: alpha-L-fucosidase (WP_254477430), 9: glycoside hydrolase family 30 (WP_254477160) ($p < 0.05$, t-test followed by Benjamini-Hochberg procedure to control for false discovery rate).

The online version of this article includes the following source data for figure 4:

**Source data 1.** SDS-PAGE gel run for cultures of *Bifidobacterium* strain wkB204.

**Source data 2.** Differential protein expression analysis for amygdalin- and glucose-grown cultures of *Bifidobacterium* strain wkB204.

carbohydrate metabolism (*Figure 4—source data 2*). Interestingly, we detected a highly expressed enzyme belonging to the glycoside hydrolase family 3 (GH3) (WP_254476944) only in amygdalin-grown cultures (*Figure 4C*), suggesting its involvement in the observed degradation. Other studies have demonstrated that specific bacterial or fungal GH3 enzymes can degrade amygdalin (*Gao and Wakarchuk, 2014*; *Guo et al., 2015*; *Chang and Zhang, 2012*; *Li et al., 2018*).

## GH3 gene expression in *Bifidobacterium* strains

We used this wkB204 GH3 (WP_254476944) as a query to search a customized database of proteins from bee gut bacteria, including 22 bee-associated *Bifidobacterium* strains. Ten other *Bifidobacterium* strains encode a GH3 in their genomes with a high sequence similarity to the wkB204 GH3 (*Figure 5—figure supplement 1*). Intriguingly, these included a GH3 from wkB344 (WP_121913979), which did not grow in the presence of amygdalin in vitro.

To determine why this GH3 does not enable wkB344 to use amygdalin as a carbon source, we investigated whether this enzyme is expressed in cultures, and used wkB204 and wkB338 cultures as controls for presence and absence of GH3 activity, respectively (*Figure 5A*). In the presence of glucose as the sole carbon source, strains wkB204 and wkB344, but not wkB338, express the GH3 gene (*Figure 5B*). When cultivated in the presence of amygdalin as the sole carbon source, only wkB204 shows elevated expression of GH3 transcripts (*Figure 5B*), which correlates with the ability of this strain to degrade amygdalin in vitro. No elevation in expression was evident for wkB344 (*Figure 5B*), and the levels of GH3 produced by wkB344 in glucose-grown cultures did not result in observable amygdalin degradation when incubated in 10 mM amygdalin (*Figure 3C*).

The wkB204 GH3 (WP_254476944), which is overexpressed in amygdalin-grown cultures, is encoded in an operon containing four other genes: a major facilitator superfamily transporter (WP_254476943), a glycoside hydrolase family 30 (WP_254477160), and a beta-galactosidase (WP_254477161) (*Figure 5C*). These were also overexpressed in the presence of amygdalin, based on proteomics data (*Figure 5C*). The wkB344 GH3 (WP_121913979) is also encoded in an operon, but a beta-galactosidase (WP_121914045) is the only other gene in the operon (*Figure 5C*), as predicted by the operon-mapper webserver (*Taboada et al., 2018*).

According to the dbCAN meta server for automated CAZyme annotation, the genomes of these three *Bifidobacterium* strains encode multiple GH3s: wkB204 encodes 10 distinct GH3s, while wkB344 and wkB338 encode 5 distinct GH3s each (*Figure 5—source data 1*; *Yin et al., 2012*; *Zhang et al., 2018*). Based on the NCBI inference database and amino acid similarity to other annotated GH3s, these three strains have some GH3s highly similar in amino acid sequence and probably similar in function (*Figure 5D* and *Table 1*), as noted for wkB204-GH3 (WP_254476944) and wkB344-GH3 (WP_121913979) (*Figure 5E*).

## *Bifidobacterium* strains also degrade prunasin

To investigate whether *Bifidobacterium* strains can also degrade prunasin, we performed an additional in vitro experiment in which *Bifidobacterium* strains wkB204, wkB344, and wkB338 were grown in 10 mM glucose in SDM in the presence of 0.1 mM prunasin (*Figure 6A*). Under these conditions, all strains grew in the presence of prunasin (*Figure 6B*) and degraded it (*Figure 6C*). For comparison, we also checked growth in the presence of 0.1 mM amygdalin (*Figure 6D*) and found that not only wkB204, but also wkB344 degraded amygdalin (*Figure 6E*). The lack of growth and, consequently, of degradation observed before for this strain is probably due to the much higher concentration of amygdalin provided in previous cultures (10 or 100 mM).

## *Escherichia coli* expressing the GH3 enzyme produces prunasin

To confirm the ability of the *Bifidobacterium* GH3 enzyme to degrade amygdalin and/or prunasin, we cloned and expressed the GH3 gene from *Bifidobacterium* strains wkB204 (WP_254476944) or wkB344 (WP_121913979) in *E. coli* (*Figure 7A–B*). Cell lysates of transformed *E. coli* expressing GH3 were incubated in the presence of 0.1 mM amygdalin or 0.1 mM prunasin (*Figure 7C*). After 5 days of incubation, we observed amygdalin degradation (*Figure 7D*) followed by prunasin production (*Figure 7E*) for *E. coli* cell lysates expressing either wkB204-GH3 or wkB344-GH3, but not for *E. coli* transformed with an empty plasmid, indicating that both enzymes can degrade amygdalin into prunasin. When the cell lysates were incubated in the presence of prunasin, only a small amount of prunasin was degraded (*Figure 7F*), suggesting that this enzyme, under the tested conditions, still can degrade prunasin, but to a lesser extent. These findings show that this *Bifidobacterium*-related GH3 enzyme can degrade amygdalin into prunasin, and potentially prunasin into mandelonitrile, and may be responsible for the degradation patterns observed for *Bifidobacterium* strain wkB204 when cultured in the presence of amygdalin.

## Host and symbionts contribute to amygdalin degradation

We also investigated amygdalin degradation in vivo. To that end, we performed experiments with bees lacking a microbiota (microbiota-deprived or MD), colonized with a conventional microbiota (CV), or monocolonized with *Bifidobacterium* strains wkB204 or wkB344 (*Figure 8A*). Bees were hand-fed 5 µL of 1 mM amygdalin in sucrose syrup, or only sucrose syrup. Amygdalin was detected in different compartments of the bee body, including the midgut, the hindgut and the body carcass

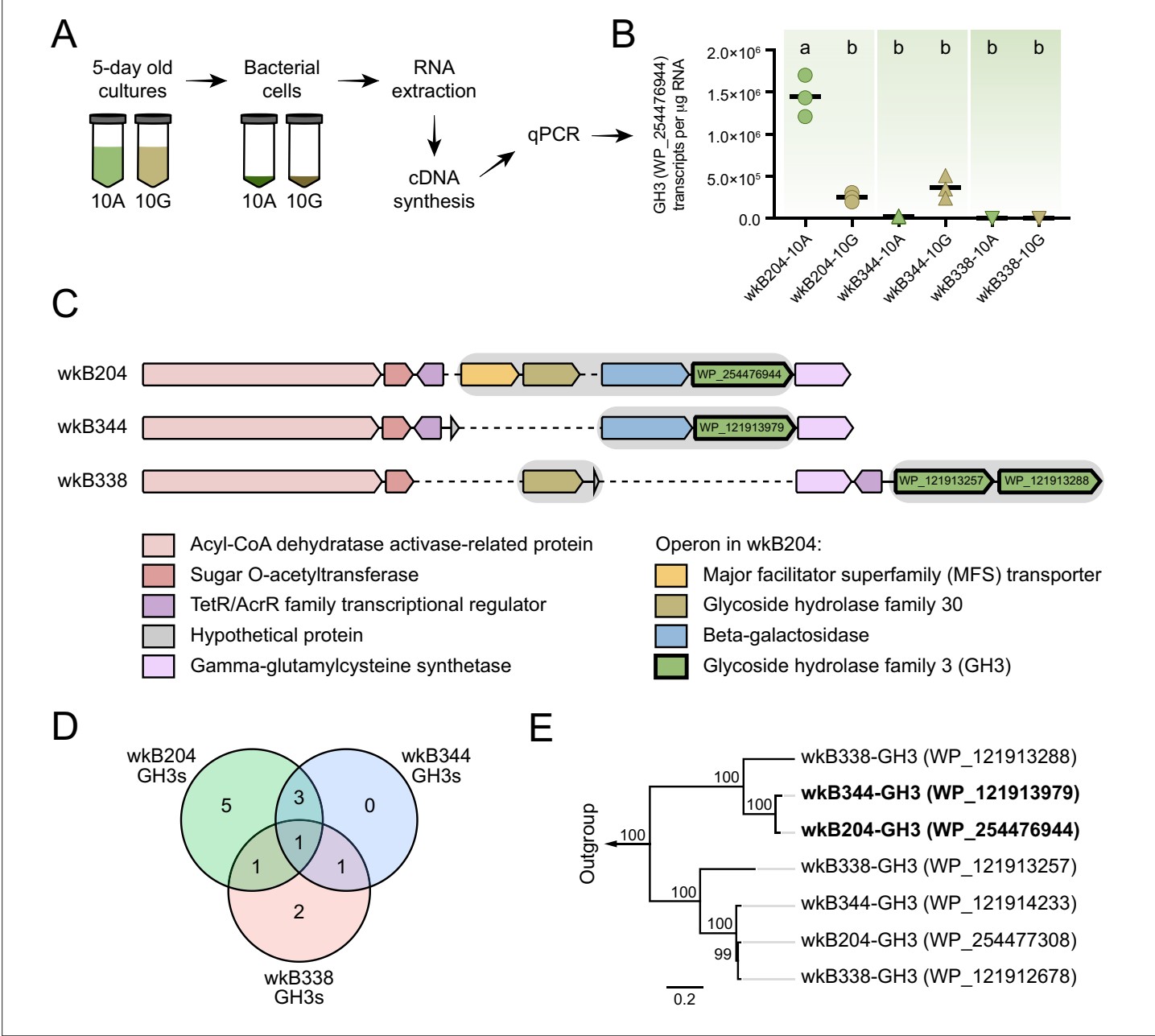

**Figure 5.** Glycoside hydrolase family 3 (GH3) gene expression in *Bifidobacterium* cultures. (**A**) RNA extraction and complementary DNA (cDNA) synthesis from cultures of *Bifidobacterium* strains wkB204, wkB344, and wkB338. (**B**) qPCR data for the transcript levels of GH3 in cells of *Bifidobacterium* strains cultured in the presence of 10 mM glucose (10G) or 10 mM amygdalin (10A). Experiments were performed in three biological replicates. Groups with different letters are significantly different (p < 0.01, one-way ANOVA test followed by Tukey's multiple-comparison test). (**C**) The genomic region containing the GH3 gene with high sequence similarity in wkB204 and wkB344. The corresponding region is included for wkB338 for comparison. Gray shading indicates operons. Dashed lines indicate regions not present in the genome. (**D**) Venn diagram showing the number of GH3s shared between the strains with amino acid similarity to other annotated GH3s according to the NCBI inference database. (**E**) Phylogenetic analysis for the GH3s found in the genomic regions shown in **C**. Outgroup is represented by two amygdalin-degrading GH3s isolated from *Rhizomucor miehei* strain RmBglu3B (AIY32164.1) and *Talaromyces cellulolyticus* strain Bgl3B (GAM39187.1).

The online version of this article includes the following source data and figure supplement(s) for figure 5:

**Source data 1.** dbCAN meta server results for *Bifidobacterium* strains wkB204, wkB344, and wkB338.

**Figure supplement 1.** Maximum-likelihood phylogeny based on amino acid sequences of bee associated *Bifidobacterium* glycoside hydrolases with sequence homology to a glycoside hydrolase family 3 highly expressed in amygdalin-grown cultures of *Bifidobacterium* strain wkB204 (PhyML 3.1, LG model + Gamma4, 100 bootstrap replicates).

**Table 1.** Glycoside hydrolases family 3 (GH3) detected in the genomes of *Bifidobacterium* strains wkB204, wkB344, and wkB338.

Protein ID refers to the unique identification of each GH3 in the NCBI Reference Sequence Database. Inference refers to the closest related GH3 present in the NCBI Reference Sequence Database. Same colors and superscript letters indicate GH3s with similar amino acid sequence. This information was used to make the Venn diagram in *Figure 5D*.

| Strain | GH3 loci number | Protein ID (NCBI RefSeq) | Inference (NCBI RefSeq) |
|---|---|---|---|
| wkB204 | 10 | WP_254476932[a] | WP_007147852 |
| | | WP_254476944[b] | WP_015021504 |
| | | WP_254477003 | WP_003842825 |
| | | WP_254477019 | – |
| | | WP_254477308[c] | WP_015022086 |
| | | WP_254477316 | – |
| | | WP_254477624[d] | WP_016461981 |
| | | WP_254477626[e] | WP_004221005 |
| | | WP_254478126 | – |
| | | WP_254478363 | – |
| wkB344 | 5 | WP_121913968[a] | WP_007147852 |
| | | WP_121913979[b] | WP_015021504 |
| | | WP_121914233[c] | WP_015022086 |
| | | WP_121914846[d] | WP_016461981 |
| | | WP_121914847 | WP_004221005 |
| wkB338 | 5 | WP_121912678[c] | WP_015022086[c] |
| | | WP_121912768 | WP_003838412 |
| | | WP_121912769[e] | WP_004221005[e] |
| | | WP_121913257 | WP_003839235 |
| | | WP_121913288 | WP_015450023 |

without the gut of MD, CV, and monocolonized bees (*Figure 8B*). In the hindgut samples, amygdalin was detected for MD and wkB344-monocolonized bees but not for CV and wkB204-monocolonized bees (*Figure 8B*). Total amygdalin concentration was significantly lower in CV bees and wkB204-monocolonized bees when compared to control bees not treated with amygdalin but spiked with 5 μL of 1 mM amygdalin during the extraction protocol (*Figure 8C*). Interestingly, prunasin was only detected in the midgut and hindgut of MD bees (*Figure 8D–E*).

These findings demonstrate the role of the microbiota in amygdalin degradation, as amygdalin concentration is reduced in CV bees and prunasin does not accumulate in the guts of CV or mono-colonized bees. These findings also show that bees themselves can degrade amygdalin, but that this degradation is partial, since prunasin accumulates in the guts of MD bees. Therefore, the presence of the microbiota contributes to continued amygdalin and prunasin degradation in the bee gut.

## Honey bees tolerate typical environmental concentrations of amygdalin

Honey bees exposed to concentrations of amygdalin, ranging from 0.01 to 1 mM, did not exhibit increased mortality rates or dysbiosis (*Figure 9A–B*). We did not find any significant changes in gut microbial composition (*Figure 9C–D*) or abundance (*Figure 9E*) of amygdalin-treated bees when compared to untreated bees, which is consistent with other studies (*Tauber et al., 2020*). More-over, amygdalin did not affect mortality rates of MD bees (*Figure 9—figure supplement 1*). The concentrations used in the in vivo experiments are below the concentrations detected in almond

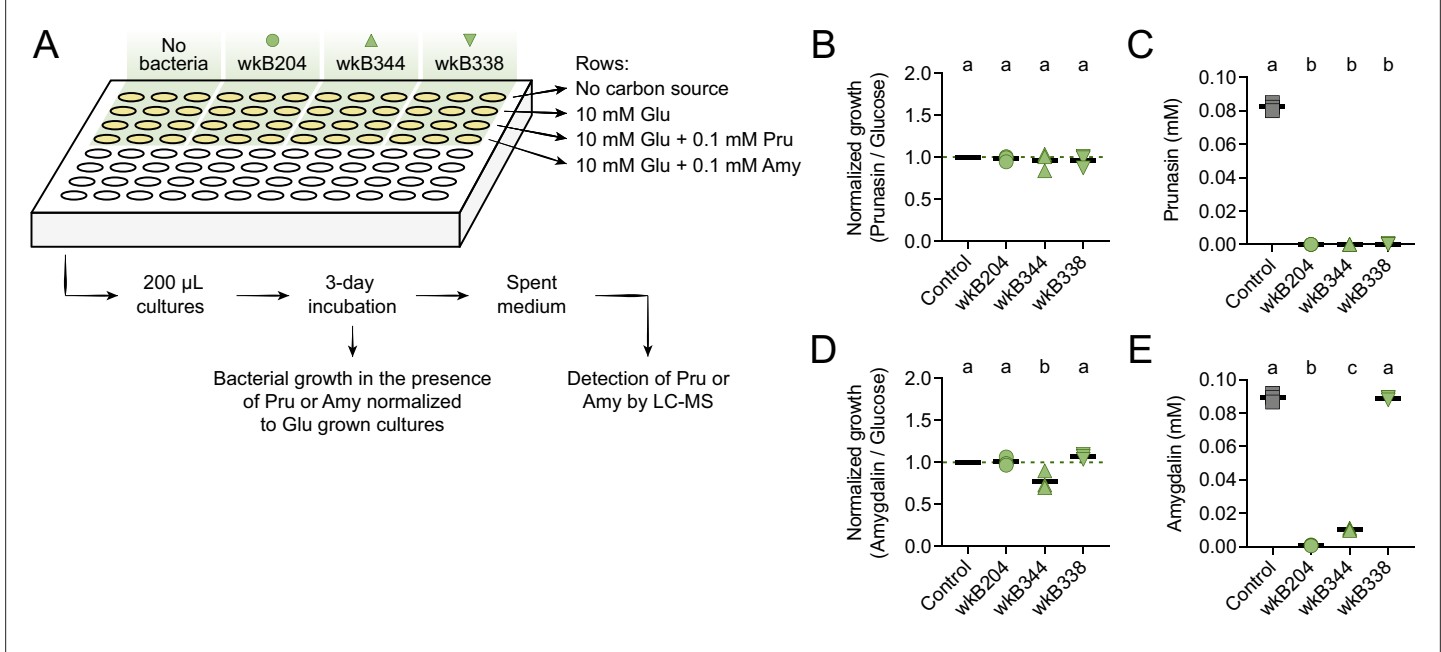

**Figure 6.** Prunasin degradation by bee gut-associated *Bifidobacterium* strains. (**A**) Experimental design. (**B**) Bacterial growth, and (**C**) prunasin degradation after 3 days of incubation in the presence of 0.1 mM prunasin. (**D**) Bacterial growth and (**E**) amygdalin degradation after 3 days of incubation in the presence of 0.1 mM amygdalin. Experiments were performed in three biological replicates. Groups with different letters are significantly different ($p < 0.01$, one-way ANOVA test followed by Tukey's multiple-comparison test).

pollen (~4 mM), and the lower concentration resembles what has been detected in almond nectar (~0.01 mM) (*London-Shafir et al., 2003*).

## Discussion

Dietary compounds can be metabolized not only by host enzymes but also by the gut microbiota, potentially resulting in increased or decreased toxicity (*Koppel et al., 2017*). Once consumed by bees, amygdalin is broken down in the bee gut, but whether this degradation is via host or pollen-derived GHs (*Pontoh and Low, 2002*; *Ricigliano et al., 2017*) or through activity of the microbiota, has been unclear. We found that members of the bee gut microbiota can degrade amygdalin and its intermediate prunasin in vitro. Specific strains of *Bifidobacterium*, *Bombilactobacillus,* and *Gilliamella* isolated from *A. mellifera*, *B. impatiens*, and *A. cerana*, respectively, were able to degrade amygdalin in vitro. In some cases, the pathway led to the production of the non-toxic intermediate prunasin; in others it did not. While the host alone can degrade amygdalin to prunasin, enzymes secreted by gut bacteria are required for the further degradation of prunasin, and the release of toxic hydrogen cyanide. Our findings for honey bees parallel those for the metabolism of amygdalin in rats, in which the molecule is degraded to produce toxic hydrogen cyanide only in the presence of the gut microbiota, but not when microbiota is absent or when injected (*Carter et al., 1980*).

Specific members of the bee gut microbiota produce a diverse set of carbohydrate digestive enzymes, including pectin lyases (PLs) and GHs, that help in food processing (*Zheng et al., 2019*; *Zheng et al., 2016*; *Engel et al., 2012*) and detoxification (*Berenbaum and Johnson, 2015*; *Koch et al., 2022*). For instance, *Gilliamella* strains produce PLs and GHs that are involved in the metabolism of pectin and hemicellulose from the pollen cell wall and toxic sugars from nectar or produced during digestion of pectin (*Zheng et al., 2016*; *Engel et al., 2012*). Some of these sugars, such as mannose, arabinose, xylose, and galactose, are indigestible for bees and can cause toxicity if accumulated in the gut (*Barker, 1977*). *Bombilactobacillus* and *Lactobacillus* strains also produce enzymes involved in mannose metabolism which potentially contribute to this detoxification mechanism (*Zheng et al., 2019*; *Ellegaard et al., 2019*). Interestingly, genomes of *Bifidobacterium* strains seem to harbor a wider repertoire of GHs than other core members of the bee gut microbiota, but lack PL-related

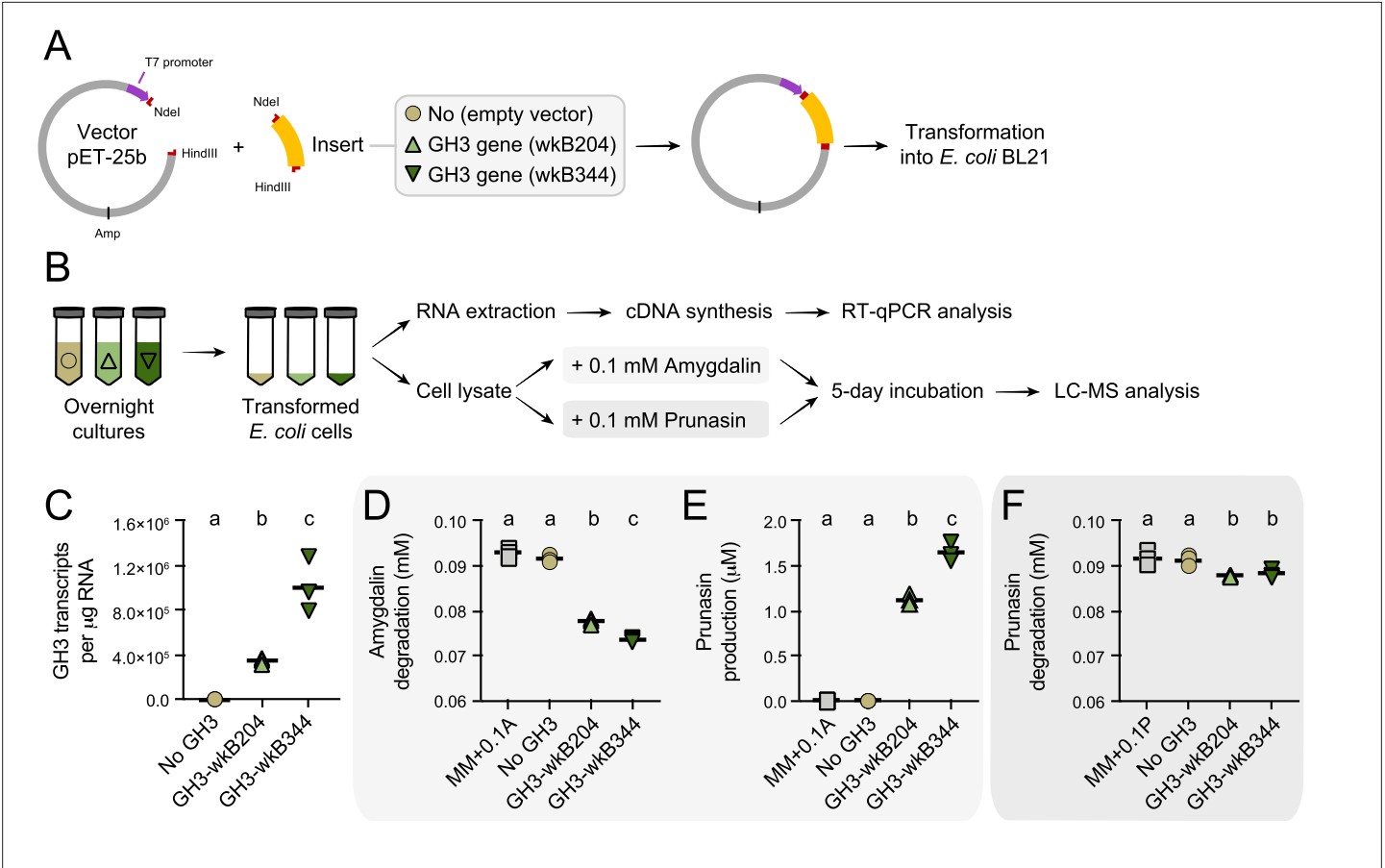

**Figure 7.** Heterologous expression of *Bifidobacterium* glycoside hydrolase family 3 (GH3) enzyme in *Escherichia coli*. (**A**) *E. coli* Rosetta BL21 competent cells were transformed with the vector pET-25b carrying the gene that encodes the wkB204-GH3 or wkB344-GH3, or only the empty vector as a control. (**B**) Bacterial cells from overnight cultures were lysed to extract RNA and investigate the expression levels of cloned genes by RT-qPCR. In parallel, bacterial cells from similar overnight cultures were lysed and used in incubation assays with 0.1 mM amygdalin or 0.1 mM prunasin in minimal medium at 37°C. Samples were submitted for LC-MS analysis along with amygdalin and prunasin standards. (**C**) Transcript levels of *Bifidobacterium*-related GH3 genes expressed in *E. coli*. (**D**) Amygdalin degradation and (**E**) prunasin production levels after 5 days of incubation in the presence of 0.1 mM amygdalin. (**F**) Prunasin degradation levels after 5 days of incubation in the presence of 0.1 mM prunasin. Experiments were performed in three biological replicates. Groups with different letters are significantly different (p < 0.01, one-way ANOVA test followed by Tukey's multiple-comparison test).

genes (*Zheng et al., 2019*). These enzymes tend to be substrate-specific and biochemical assays are usually required to verify function. In our study, we identified a specific GH3 in *Bifidobacterium* strains that contributes to amygdalin and prunasin metabolism in vitro. Other studies have shown that bacterial- or fungal-derived GH3s can degrade amygdalin. For example, the Gram-positive bacterium *Cellulomonas fimi* encodes a GH3 with activity against β-1,6-linked glycosides (*Gao and Wakarchuk, 2014*), similar to the linkage found in the structure of amygdalin (*Figure 2D*). Degradation of amygdalin by GH3s isolated from *Rhizomucor miehei* (*Guo et al., 2015*) and *Talaromyce leycettanus* (*Li et al., 2018*), or by related extracellular enzymes from *Aspergillus niger*, has also been observed (*Chang and Zhang, 2012*). Moreover, different species of mammalian gut-associated *Bifidobacterium* strains can grow in the presence of amygdalin, potentially due to the production of GH1 or GH3 enzymes (*Modrackova et al., 2020*).

In our case, heterologous expression of wkB204-GH3 (WP_254476944) or wkB344-GH3 (WP_121913979) using *E. coli* also led to amygdalin degradation, but to a lesser extent than what was observed for the original host. To accurately quantify the lower degradation rates in transformed *E. coli*, we used 0.1 mM amygdalin solutions (*Figure 7*). Potentially, there are other carbohydrate digestive enzymes encoded by *Bifidobacterium* that contribute to amygdalin or prunasin metabolism (*Figure 4C*), or the *Bifidobacterium* host perform specific posttranslational modifications on this

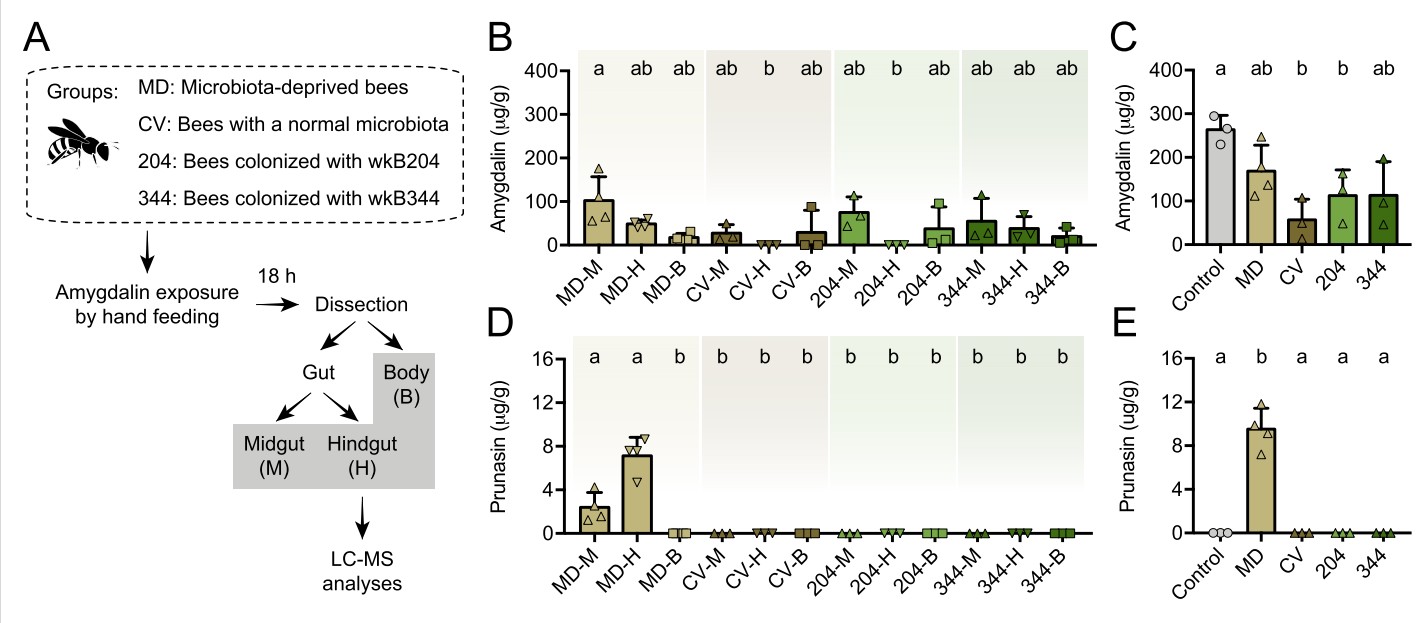

**Figure 8.** Amygdalin metabolism in honey bees. (**A**) Five-day old bees either lacking a microbiota (microbiota deprived, MD, n=4), with a normal microbiota (conventionalized, CV, n=3), or monocolonized with *Bifidobacterium* strains wkB204 (n=3) or wkB344 (n=3), were exposed to 5 µL of 1 mM amygdalin and dissected 24 hr later to determine the concentrations of (**B**) amygdalin and (**D**) prunasin in different bee body compartments (midgut: M, hindgut: H, and body without gut: B) by LC-MS. (**C**) Amygdalin and (**E**) prunasin concentrations detected in M, H, and B samples were summed for each group and compared to a control group of unexposed bees that were mixed with 5 µL of 1 mM amygdalin at the beginning of sample processing. Groups with different letters are significantly different (p < 0.05, one-way ANOVA test followed by Tukey's multiple-comparison test).

enzyme that are not achieved during heterologous expression in *E. coli* (*Adav et al., 2014*). Moreover, it seems this GH3 is secreted by *Bifidobacterium* strain wkB204, but, for cloning purposes, we expressed it in *E. coli* without a signal sequence for secretion, and, therefore, performed assays with cell lysates, which may not be optimal for characterizing enzyme activity. These features may have masked the potential activity of this GH3, if this is the main enzyme involved in amygdalin degradation by *Bifidobacterium* strains. Moreover, other hydrolases were identified in the proteomics data (*Figure 4C*). Although not reported in the literature, some of these could potentially be involved in amygdalin degradation.

As shown in other studies, nectar and pollen metabolites can be chemically transformed when passing through the gut, and this may influence their effects on the host and/or the microbiota (*Koch et al., 2019*; *Vidkjær et al., 2021*). To identify the potential contribution of both the host and the microbiota to amygdalin degradation, we performed in vivo experiments in which MD bees and microbiota-colonized bees were exposed to amygdalin for a short period of time. We found that MD bees can degrade amygdalin, but only partially, leading to accumulation of prunasin in gut compartments. In contrast, prunasin accumulation was not observed in microbiota-colonized bees or in bees monocolonized with specific *Bifidobacterium* strains, and amygdalin degradation was higher in these groups when compared to MD bees. This suggests that members of the microbiota, besides contributing to amygdalin degradation, can also efficiently degrade prunasin and potentially release the final products of amygdalin metabolism, such as hydrogen cyanide. This could potentially increase the side effects of amygdalin byproducts on bees or on the microbiota, since hydrogen cyanide can be toxic to aerobic organisms (*Khandekar and Edelman, 1979*; *Hurst et al., 2014*). However, similar to other studies, we did not detect increased mortality rates or changes in microbial community abundance and composition for bees exposed to amygdalin (*Lecocq et al., 2018*; *Tauber et al., 2020*), strongly suggesting that field-relevant concentrations of amygdalin are not detrimental to bees. Interestingly, we also found amygdalin in bee carcasses with the gut removed, suggesting that amygdalin is absorbed systemically by bee cells. This has been observed in other studies in which amygdalin was found in the bee hemolymph after oral ingestion (*Hurst et al., 2014*; *Vidkjær et al., 2021*).

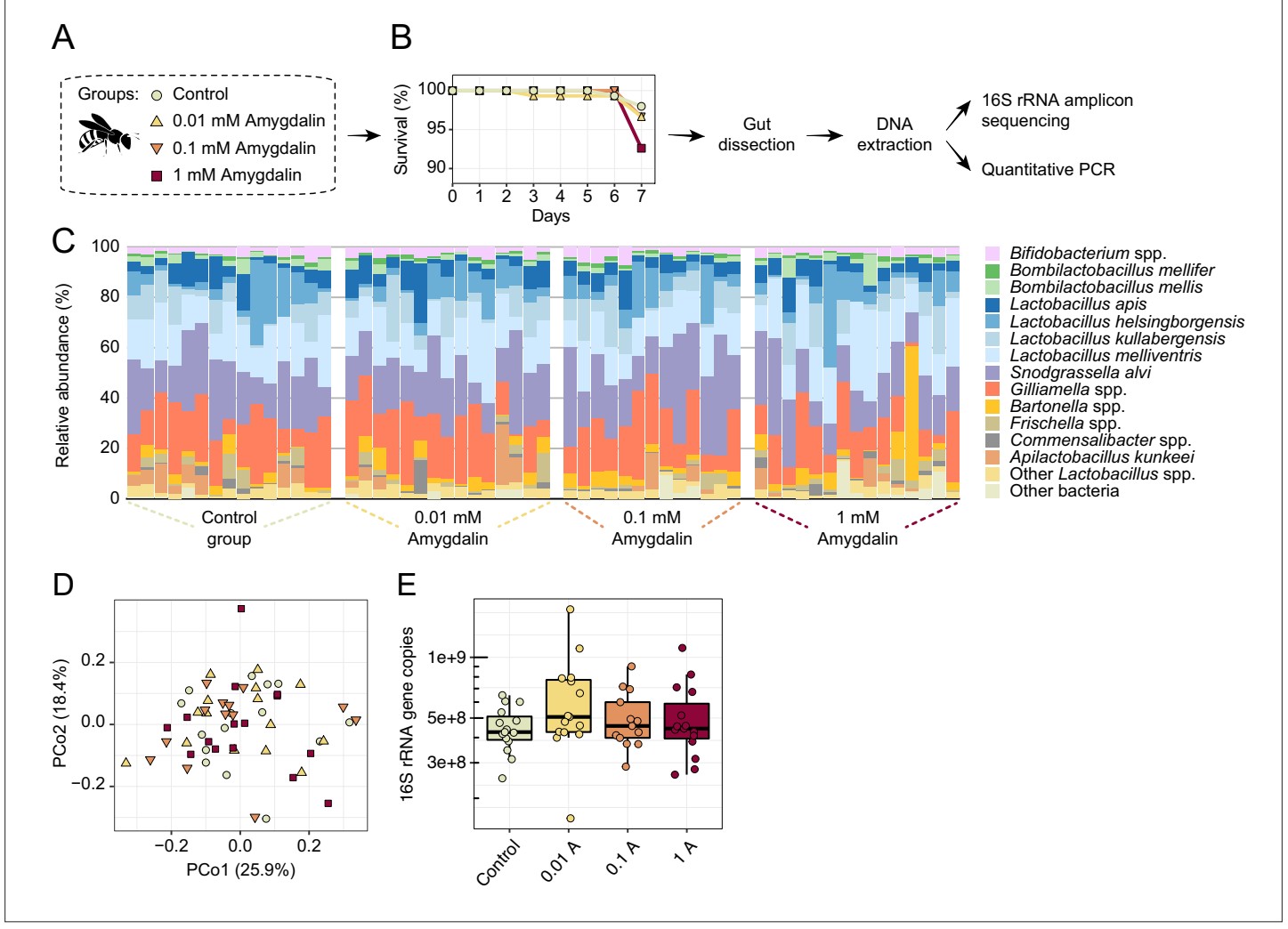

**Figure 9.** Amygdalin effects on the honey bee gut microbiota. (**A**) Experimental design and (**B**) survival rates of honey bees exposed to different concentrations of amygdalin. (**C**) Stacked column graphs showing the relative abundance of bee gut bacterial species in control bees (n=15), 0.01 mM amygdalin (n=15), 0.1 mM amygdalin (n=13), and 1 mM amygdalin (n=15) exposed bees. (**D**) Principal coordinate analysis of gut community compositions of control and amygdalin exposed bees using Bray-Curtis dissimilarity (p>0.5, Permanova test with 9999 permuations). (**E**) Boxplot of total bacterial 16S rRNA gene copies estimated by qPCR for control and amygdalin exposed bees. Box-and-whisker plots show high, low, and median values, with lower and upper edges of each box denoting first and third quartiles, respectively. No significant differences were observed in total bacterial abundance between control and amygdalin exposed bees (p>0.05, Kruskal-Wallis test).

The online version of this article includes the following figure supplement(s) for figure 9:

**Figure supplement 1.** Survival rates of honey bees exposed to different concentrations of amygdalin.

Studies in other animals have also investigated the roles of the microbiota on amygdalin degradation. Studies in rats, for example, have found higher concentrations of hydrogen cyanide in the blood of microbiota-colonized rats than in MD rats (*Carter et al., 1980*) or antibiotic-treated rats (*Newton et al., 1981*) after oral ingestion of amygdalin. However, intravenous administration of amygdalin seems not to lead to hydrogen cyanide formation (*Jaswal and Palanivelu, 2018*), which could be correlated to the lack of toxicity observed in honey bees after amygdalin injection into the hemolymph (*Hurst et al., 2014*). These studies suggest that the gut microbiota is a major factor driving amygdalin degradation and hydrogen cyanide release in the guts of animals.

It is important to note that colonization by the normal microbiota decreases the pH to about 5 in the bee gut (*Zheng et al., 2017*), and low pH can favor the degradation of amygdalin into prunasin, then prunasin into mandelonitrile, which can undergo spontaneous degradation at acidic pH to give benzaldehyde and hydrogen cyanide (*Jaswal and Palanivelu, 2018*). Therefore, the presence of the

microbiota itself, without the action of GHs, could favor the degradation of amygdalin into prunasin and potentially lead to the production of hydrogen cyanide.

Although bees and their native microbiota seem to tolerate relatively high doses of amygdalin, parasites that commonly inhabit the bee gut may not fare as well. There is increasing evidence that metabolites in nectar and pollen, even those considered toxic in some cases, can improve pollinator health at specific concentrations by controlling or reducing parasite loads (*Stevenson et al., 2017*; *Tauber et al., 2020*; *Costa et al., 2010*; *Simone-Finstrom and Spivak, 2012*; *Stevenson, 2020*). Indeed, bees tend to forage on specific plants as a means of reducing colony pathogen loads (*Simone-Finstrom and Spivak, 2012*). For example, honey bees from hives treated with amygdalin exhibited decreased levels of infection by the parasite *L. passim* and some viruses (*Tauber et al., 2020*). In contrast, this does not seem to be the case for *Crithidia* infection in bumble bees, whose loads are not reduced after amygdalin exposure (*Richardson et al., 2015*), as the parasite is not susceptible to amygdalin (*Palmer-Young et al., 2016*). Other plant metabolites, such as the essential oil thymol, can reduce *Nosema* spore loads in honey bees (*Costa et al., 2010*; *Stevenson, 2020*). Therefore, the extent of protection may depend on the exposure level and on the parasite being exposed.

Indeed, some insects, such as bees, ants, flies, and butterflies, can use a wide range of toxic secondary metabolites to medicate themselves in a therapeutic or prophylactic way (*de Roode and Hunter, 2019*). Honey bees typically collect secondary metabolites in plant resins, nectar, and pollen as prophylactic medication behavior to protect their colonies from parasites (*Simone-Finstrom and Spivak, 2012*; *Erler and Moritz, 2016*; *Simone-Finstrom et al., 2017*). Similarly, Monarch butterflies feed on milkweeds with high concentrations of toxic cardenolides as a therapeutic medication against parasites (*Gowler et al., 2015*). Monarchs can also use cardenolides as a prophylactic medication by selecting oviposition sites with high concentrations of cardenolides (*Lefèvre et al., 2012*) or transferring cardenolides to eggs (*Sternberg et al., 2015*) to reduce parasite growth in their hatching offspring. Some species of fruit flies use transgenerational medication to resist attack from parasitoid wasps (*Poyet et al., 2017*). Female flies preferentially oviposit in media containing the alkaloid atropine, which reduces infection success of parasitoids, but also reduces fecundity.

More recently, a few studies have brought attention to activities of plant-derived compounds within the host gut, which can be modulated by the host and/or the microbiota, to lead to the final metabolite activity (*Koch et al., 2019*). The recently described plant metabolite callunene, from heather nectar, can play a prophylactic role in preventing *Crithidia* infections in bumble bees, especially when parasite cells are present in the crop. However, callunene cannot play a therapeutic role in infected hosts, because host metabolism inactivates it before it reaches the hindgut where *Crithidia* usually establishes (*Koch et al., 2019*).

Not only the host, but also the microbiome can play a major role in the metabolism of some nectar metabolites with consequences for activity against parasites. The glycosylation status of a nectar metabolite can directly affect its activity, with the aglycones exhibiting higher activity than the corresponding glycosylated metabolite. This has been observed for the metabolites tiliaside, from linden trees, and unedone, from strawberries (*Koch et al., 2022*). Tiliaside is a glycoside with in vivo, but not in vitro, activity against *Crithidia bombi* as it requires deglycosylation by host and/or microbiome enzymes during gut passage to exhibit antiparasitic activity (*Koch et al., 2022*). Unedone, on the other hand, is not a glycoside and has both in vitro and in vivo activity against *C. bombi*. Interestingly, unedone is first glycosylated and inactivated in the midgut by bee enzymes, then deglycosylated and reactivated in the hindgut by the microbiome (*Koch et al., 2022*).

A few other studies have also investigated the roles of the microbiota in the metabolism of other plant metabolites. *Kešnerová et al., 2017*, demonstrated that honey bees monocolonized with strains of *Bifidobacterium*, *Bombilactobacillus*, or *Lactobacillus* can metabolize flavonoid glycosides. Indeed, genomic analyses have shown that the genomes of strains from these bacterial groups contain a diverse set of carbohydrate processing genes, including GHs that could be involved in the cleavage of sugar residues (*Zheng et al., 2019*; *Ellegaard et al., 2019*). We found strain variation for amygdalin metabolism, and such variation likely influences processing of other dietary secondary metabolites. Strain-level variations in the microbiome could, consequently, affect parasite persistence or establishment. Perturbation of the honey bee gut microbiota by pesticides and/or heavy metals (*Motta and Moran, 2020b*; *Rothman et al., 2020*; *Motta et al., 2022b*) could indirectly affect parasite success in the bee gut through changes in the metabolism of secondary metabolites by the microbiome.

Interactions between gut microbiomes and pathogens are widespread and have repeatedly been shown for the bee gut microbiota (*Raymann and Moran, 2018*). In honey bees, the core microbiota contributes to protection against RNA viruses (*Dosch et al., 2021*), pathogenic bacterial infections (*Steele et al., 2021*; *Lang et al., 2022*), and the microsporidian parasite *Nosema* (*Wu et al., 2020*). Further, infection by *Nosema,* for example, can lead to microbiome dysbiosis (*Paris et al., 2020*; *Rubanov et al., 2019*; *Huang et al., 2018*). Although microbiota-conferred protection seems to be common in bees, the underlying mechanisms are generally unknown. They may involve the host immune system (*Horak et al., 2020*; *Kwong et al., 2017a*) and/or direct microbial interactions within the gut (*Steele et al., 2017*). Our study on amygdalin shows that microbial metabolism of dietary components may be one mechanism through which microbiota members impact hosts or co-resident microbes.

Our results show the relevance of the microbiota for the metabolism of plant toxins, using amygdalin as an example of a toxin that is jointly metabolized by the host and the microbiota. The consequences for bees of the full metabolism of amygdalin are not yet clear-cut but point toward the beneficial to neutral spectrum of host-microbe interactions. The metabolism of most field-relevant concentrations of amygdalin does not affect bee survival rates or microbiota composition and does not deter bees (*Tiedeken et al., 2014*). In some instances, dietary amygdalin can reduce parasite and viral loads (*Tauber et al., 2020*; *Palmer-Young et al., 2017*). Future work could investigate the amounts of hydrogen cyanide released during in vitro experiments with amygdalin-degrading bacteria or during in vivo experiments with microbiota-colonized and microbiota-deprived bees. The full impact of amygdalin on bee health could be assessed using experiments testing whether hydrogen cyanide released into the gut may protect bees infected by specific parasites. Such experimental studies could better elucidate how gut microbial communities metabolize plant metabolites and how this metabolism affects host fitness.

# Methods

## Key resources table

| Reagent type (species) or resource | Designation | Source or reference | Identifiers | Additional information |
|---|---|---|---|---|
| Gene (*Bifidobacterium asteroides*) | wkB204 | NCBI Reference Sequence | Locus: WP_254476944 | |
| Gene (*Bifidobacterium asteroides*) | wkB344 | NCBI Reference Sequence | Locus: WP_121913979 | |
| Strain, strain background (*Escherichia coli*) | DH5-alpha | New England BioLabs | Cat#: C2987H | NEB 5-alpha competent cells |
| Strain, strain background (*Escherichia coli*) | BL21 (DE3) | New England BioLabs | Cat#: C2527H | Electrocompetent cells |
| Strain, strain background (*Bifidobacterium asteroides*) | wkB204 | This paper | JAFMNU020000000 | Bacterial isolate |
| Strain, strain background (*Bifidobacterium asteroides*) | wkB344 | doi:10.1073/pnas.1916224116 | NPOQ00000000 | Bacterial isolate |
| Strain, strain background (*Bifidobacterium asteroides*) | wkB338 | doi:10.1073/pnas.1916224116 | NPOR00000000 | Bacterial isolate |
| Strain, strain background (*Bombilactobacillus bombi*) | BI-2.5 | This paper | CP031513 | Bacterial isolate |
| Strain, strain background (*Bombilactobacillus bombi*) | BI-1.1 | This paper | QOCR00000000 | Bacterial isolate |
| Strain, strain background (*Bombilactobacillus bombi*) | LV-8.1 | This paper | QOCS00000000 | Bacterial isolate |
| Strain, strain background (*Bombilactobacillus mellifer*) | Bin4N | doi:10.1099/ijs.0.059600–0 doi:10.1099/ijsem.0.004107 | JXJQ00000000 | Bacterial isolate |
| Strain, strain background (*Lactobacillus bombicola*) | OCC3 | This paper | QOCV00000000 | Bacterial isolate |
| Strain, strain background (*Lactobacillus bombicola*) | BI-4G | This paper | QOCU00000000 | Bacterial isolate |
| Strain, strain background (*Lactobacillus* nr. *melliventris*) | HB-1 | This paper | OQ216581 | Bacterial isolate |

*Continued on next page*

*Continued*

| Reagent type (species) or resource | Designation | Source or reference | Identifiers | Additional information |
|---|---|---|---|---|
| Strain, strain background (*Lactobacillus* nr. *melliventris*) | HB-2 | This paper | OQ216582 | Bacterial isolate |
| Strain, strain background (*L.* nr. *melliventris*) | HB-C2 | This paper | OQ216583 | Bacterial isolate |
| Strain, strain background (*Lactobacillus* nr. *melliventris*) | HB-D10 | This paper | OQ216584 | Bacterial isolate |
| Strain, strain background (*Lactobacillus helsingborgensis*) | wkB8 | doi:10.1128/genomeA.01176–14 | CP009531 | Bacterial isolate |
| Strain, strain background (*Lactobacillus kullabergensis*) | wkB10 | doi:10.1128/genomeA.01176–14 | JRJB00000000 | Bacterial isolate |
| Strain, strain background (*Gilliamella apicola*) | wkB1 | doi:10.1073/pnas.1405838111 | CP007445 | Bacterial isolate |
| Strain, strain background (*Gilliamella apicola*) | wkB7 | doi:10.1128/mBio.01326–16 | LZGG00000000 | Bacterial isolate |
| Strain, strain background (*Gilliamella apis*) | M1-2G | doi:10.1128/mBio.01326–16 | LZGQ00000000 | Bacterial isolate |
| Strain, strain background (*Gilliamella* sp.) | wkB112 | doi:10.1128/mBio.01326–16 | LZGL00000000 | Bacterial isolate |
| Strain, strain background (*Gilliamella* sp.) | wkB178 | doi:10.1128/mBio.01326–16 | LZGK00000000 | Bacterial isolate |
| Strain, strain background (*Gilliamella* sp.) | wkB108 | doi:10.1128/mBio.01326–16 | LZGM00000000 | Bacterial isolate |
| Strain, strain background (*Gilliamella* sp.) | wkB308 | doi:10.1128/mBio.01326–16 | LZGN00000000 | Bacterial isolate |
| Strain, strain background (*Gilliamella* sp.) | M6-3G | doi:10.1128/mBio.01326–16 | MCIU00000000 | Bacterial isolate |
| Biological sample (*Apis mellifera*) | Western honey bee *Apis mellifera* | Collected from hives at UT-Austin | | |
| Recombinant DNA reagent | pGEM-T Easy vector (plasmid) | Promega | Cat#: A1360 | |
| Recombinant DNA reagent | pET25b (plasmid) | Novagen | Cat#: 69753 | |
| Recombinant DNA reagent | pET25b-wkB204-GH3 (plasmid) | This study | | pET25b expressing wkB204-GH3 (WP_254476944) |
| Recombinant DNA reagent | pET25b-wkB344-GH3 (plasmid) | This study | | pET25b expressing wkB344-GH3 (WP_121913979) |
| Sequence-based reagent | B-GH3-F | This paper | PCR primers | ctaccgcaatcccgacct |
| Sequence-based reagent | B-GH3-R | This paper | PCR primers | cacctccttgtccactccc |
| Sequence-based reagent | GH3-NdeI-F | This paper | PCR primers | ttgtttaactttaagaaggagatatacatatggcat caaggaagttgacagagg |
| Sequence-based reagent | GH3-HindIII-R | This paper | PCR primers | agcccgtttgatctcgagtgcggccgcaa gcttacccacggtcaccgtca |
| Commercial assay or kit | Quick-RNA Miniprep kit | Zymo Research | Cat#: R1055 | |
| Commercial assay or kit | iTaq Universal SYBR Green Supermix | Bio-Rad | Cat#: 172–5125 | |
| Commercial assay or kit | Monarch Plasmid Miniprep Kit | New England BioLabs | Cat#: T1010L | |
| Commercial assay or kit | qScript cDNA Synthesis Kit | QuantBio | Cat#: 95047–500 | |
| Chemical compound, drug | Amygdalin | Chem-Impex International | Cat#: 22029 | Lot#: 002681–16112001 |
| Chemical compound, drug | Prunasin | Toronto Research Chemicals | Cat#: P839000 | Lot#: 6-EQJ-155–1 |
| Chemical compound, drug | Ampicillin | Fisher Bioreagents | Cat#: BP1760-5 | |
| Chemical compound, drug | Isopropyl β-D-1-thiogalactopyranoside (IPTG) | Gold Biotechnology | Cat#: I2481C25 | |
| Chemical compound, drug | Antarctic Phosphatase | New England BioLabs | Cat#: M0289S | Enzyme |
| Chemical compound, drug | NdeI | New England BioLabs | Cat#: R0111S | Restriction enzyme |

*Continued on next page*

*Continued*

| Reagent type (species) or resource | Designation | Source or reference | Identifiers | Additional information |
|---|---|---|---|---|
| Chemical compound, drug | HindIII-HF | New England BioLabs | Cat#: R3104S | Restriction enzyme |
| Software, algorithm | SeaView | http://pbil.univ-lyon1.fr/software/seaview3.html | RRID:SCR_015059 | |
| Other | Insectagro DS2 media | Corning | Cat#: 13-402-CV | Lot#: 12818007 |
| Other | Difco Lactobacilli MRS broth | BD | Cat#: 288130 | Lot#: 9211338 |
| Other | Heart Infusion Agar | Criterion | Cat#: C5822 | Lot#: 491030 |
| Other | Defibrinated Sheep Blood | HemoStat Laboratories | Cat#: DSB1 | Lot#: 663895–2 |
| Other | Protein extraction reagent (B-PER) | Thermo Scientific | Cat#: 78248 | Lot#: LJ148147A |
| Other | Bolt 4–12% Bis-Tris Plus Gel | Thermo Scientific | Cat#: NW04120BOX | Lot#: 21022470 |

## Chemicals, media, and solutions

Amygdalin was obtained from Chem-Impex International, Inc (catalog number: 22029, lot number: 002681-16112001). Prunasin was obtained from Toronto Research Chemicals, Inc (catalog number: P839000, lot number: 6-EQJ-155-1). An SDM (for recipe see *Table 2* and *Menon et al., 2013*) was used to culture *Bifidobacterium* and *Bombilactobacillus* strains. The nutrient-rich medium Insectagro DS2 (Corning, Inc, catalog number: 13-402-CV, lot number: 12818007) was used to culture *Gilliamella* strains. The Difco Lactobacilli MRS broth (BD, Inc, catalog number: 288130, lot number: 9211338) was used to culture *Lactobacillus* strains. Luria-Bertani (LB) or a minimal medium (MM, for recipe see *Table 3* and *Li et al., 2014*) was used to culture transformed *E. coli* strains. For experiments with bacterial isolates, a 1 M amygdalin solution was prepared by dissolving 4.57 g amygdalin in 10 mL of culture medium, then diluted to final concentrations of 0.1, 1, 10, or 100 mM in the same culture medium.

**Table 2.** Composition of a semi-defined medium (SDM) recipe used to culture *Bifidobacterium* and *Lactobacillus* strains.
Specific carbon sources (amygdalin and/or glucose) were added according to the experiments. Recipe was adapted from *Walker et al., 2014*.

| Ingredient | Amount (g/L) |
|---|---|
| **Defined** | |
| Ammonium chloride | 2 |
| Cysteine hydrochloride | 0.4 |
| Magnesium chloride | 0.08 |
| Manganese chloride | 0.08 |
| Nicotinic acid | 0.5 |
| Pantothenic acid | 0.5 |
| Potassium phosphate monobasic | 2 |
| Pyridoxine hydrochloride | 0.1 |
| Sodium acetate | 5 |
| **Undefined** | |
| Yeast extract | 4 |
| SC, synthetic complete supplement (Sunrise Scientific Products, Knoxville, TN, USA) | 2 |
| Tween 80 | 1 |

Also, a 5 mM prunasin solution was prepared by dissolving 5 mg prunasin in 3387 µL sterile water, then an aliquot was transferred to SDM or MM to a final concentration of 0.1 mM prunasin. For experiments with honey bees, a 10 mM amygdalin solution was prepared by dissolving 45.74 mg amygdalin in 10 mL sterile water, then diluted to final concentrations of 0.01 , 0.1, or 1 mM with filter-sterilized 0.5 M sucrose syrup and provided to bees in cup cages.

## Isolation and characterization of *Bifidobacterium* strains

*Bifidobacterium* strains wkB204, wkB344, and wkB338 were isolated from fresh guts of *A. mellifera* workers from hives kept at UT-Austin (August 2014). Guts were homogenized in 10% PBS and cultured on Heart Infusion agar at 35°C and 5% $CO_2$ for 3–5 days. Genomic DNA was extracted from overnight cultures, as in *Kwong et al., 2017b*. The wkB204 genome was sequenced on the Illumina MiSeq platform from 2×150 bp paired-end libraries at the SeqCenter (Pittsburgh, PA) and assembled using CLC Genomics Workbench 5.5 (QIAGEN). The wkB344 and wkB338 genomes were previously reported in *Zheng et al., 2019*.

**Table 3.** Composition of a minimum medium (MM, pH 6.8) recipe used to culture transformed *Escherichia coli* strains.

Specific carbon sources (amygdalin, prunasin, or glucose) were added according to the experiments. Recipe was adapted from *Li et al., 2014*.

| Ingredient | Amount (g/L) |
|---|---|
| Ammonium iron (III) citrate | 0.1 |
| Ammonium phosphate tetrahydrate | 4 |
| Boric acid | 0.003 |
| Citric acid | 1.55 |
| Cobalt (II) chloride hexahydrate | 0.003 |
| Copper (II) chloride dihydrate | 0.002 |
| Magnesium sulfate | 0.59 |
| Manganese (II) chloride tetrahydrate | 0.015 |
| Potassium phosphate monobasic | 13.3 |
| Sodium molybdate dihydrate | 0.002 |
| Zinc sulfate heptahydrate | 0.034 |

## Isolation and characterization of *Bombilactobacillus* and *Lactobacillus* strains

Bee gut-associated bacterial strains were isolated from fresh guts of commercial (strains BI-2.5, BI-1.1) and wild-caught (strain BI-4G) *B. impatiens* workers, preserved guts of *B. appositus* (strain LV-8.1) and *B. occidentalis* (strain OCC3) workers. Wild *B. impatiens* were collected in New Haven, CT, USA (August 2013); *B. appositus* and *B. occidentalis* were collected in Logan, UT, USA (July 2013); commercial *B. impatiens* were obtained from BioBest (Romulus, MI, USA). Guts and feces were homogenized in 10% PBS and cultured in MRS broth at 35°C and 5% $CO_2$ for 3–5 days, then plated on MRS agar and incubated at 35°C and 5% $CO_2$. Several passages on MRS agar were required to achieve pure isolates. Genomic DNA was extracted from overnight cultures, as in *Kwong et al., 2017b*.

The BI-2.5 genome was sequenced and closed using Pacific Biosciences technology at the Yale Center for Genome Analysis. Indel errors were corrected with Pilon using Illumina MiSeq reads – 150 bp single-read libraries sequenced at the GSAF, UT-Austin (*Walker et al., 2014*). The other four genomes were sequenced on the Illumina MiSeq platform from 2×300 bp paired-end libraries at the GSAF, UT-Austin, and assembled using CLC Genomics Workbench 5.5 (QIAGEN). All genomes were annotated with the Rapid Annotation using Subsystem Technology (RAST) server (*Overbeek et al., 2014*). Strains BI-2.5, BI-1.1, and LV-8.1 are most related to *Bombilactobacillus bombi* (*Zheng et al., 2020*; *Killer et al., 2014*), while strains OCC3 and BI-4G are most related to *Lactobacillus bombicola* (*Praet et al., 2015*; *Wang et al., 2018*).

*Bombilactobacillus mellifer* strain Bin4N (DSM 26254) was obtained from the Leibniz Institute, Germany, and is a honey bee isolate (*Zheng et al., 2020*; *Olofsson et al., 2014*).

*Lactobacillus* strains HB-1, HB-2, HB-C2, and HB-D10 were isolated from fresh guts of *A. mellifera* workers from hives kept at UT-Austin (August 2017). Guts were homogenized in 10% PBS and cultured in MRS broth at 35°C and 5% $CO_2$ for 3–5 days. Aliquots of bacterial cultures were plated on MRS agar and incubated at 35°C and 5% $CO_2$. Several passages on MRS agar were required to achieve pure isolates. Sequencing the 16S rRNA gene showed that these isolates corresponded to the bee-restricted cluster that contains *Lactobacillus melliventris*.

*Lactobacillus* strains wkB8 and wkB10 were previously isolated from the guts of *A. mellifera* (*Kwong et al., 2014*).

### Isolation and characterization of *Gilliamella* strains

*Gilliamella* strains were previously isolated from the guts of *A. dorsata* (wkB112, wkB178, wkB108), *A. cerana* (wkB308), or *A. mellifera* (M6-3G, M1-2G, wkB7, wkB1) (*Zheng et al., 2016*).

### Exposure of bee gut bacteria to amygdalin

All strains were initially cultured in Heart Infusion Agar (Criterion, New York, NY, USA, catalog number: C5822, lot number: 491030) with 5% Defibrinated Sheep Blood (HemoStat Laboratories, Dixon, CA, USA, lot number: 663895-2) at 35°C and 5% $CO_2$ for 3–5 days, then colonies were transferred to proper liquid media to obtain enough bacterial mass for in vitro experiments.

Strains of *Bifidobacterium* (wkB204, wkB338, wkB344) and *Bombilactobacillus* (BI-1.1, BI-2.5, LV-8.1, Bin4N) were cultured in SDM (*Table 2*) at 35°C and 5% $CO_2$ overnight. Optical density (OD) of each bacterial culture was measured at 600 nm, and the cells were diluted to an OD of 0.5 in SDM.

Ten µL aliquots of each bacterial suspension were transferred in three biological replicates to 96-well plates containing 190 µL SDM with no carbon sources, 10- or 100 mM amygdalin, 10- or 100 mM glucose, or 10- or 100 mM amygdalin and glucose as carbon sources. Controls consisted of three biological replicates of 200 µL SDM with similar carbon sources, but without bacterial suspension. The plates were incubated in a plate reader (Tecan) at 35°C and 5% $CO_2$, and OD was measured at 600 nm after 72 hr.

Strains of *Gilliamella* (wkB112, wkB178, wkB108, wkB308, M6-3G, M1-2G, wkB7, wkB1) were cultured in a nutrient-rich medium, Insectagro DS2 (Corning Inc), at 35°C and 5% $CO_2$ overnight. OD of each bacterial culture was measured at 600 nm, and the cells were diluted to an OD of 0.5 in Insectagro. Ten µL aliquots of each bacterial suspension were transferred in three biological replicates to 96-well plates containing 190 µL Insectagro with 10- or 100 mM amygdalin, or without amygdalin. Controls consisted of three biological replicates of 200 µL Insectagro with similar carbon sources, but without bacterial suspension. The plates were incubated in a plate reader (Tecan) at 35°C and 5% $CO_2$ and OD was measured at 600 nm after 72 hr.

Strains of *Lactobacillus* nr. *melliventris* (HB-1, HB-2, HB-C2, HB-D10, wkB8, wkB10, BI-4G, OCC3) were cultured in MRS broth at 35°C and 5% $CO_2$ overnight. OD of each bacterial culture was measured at 600 nm, and the cells were diluted to an OD of 0.5 in MRS. Ten µL aliquots of each bacterial suspension were transferred in three biological replicates to 96-well plates containing 190 µL MRS with 10 mM amygdalin, or without amygdalin. Controls consisted of three biological replicates of 200 µL MRS with similar carbon sources, but without bacterial suspension. The plates were incubated in a plate reader (Tecan) at 35°C and 5% $CO_2$ and OD was measured at 600 nm after 72 hr.

At the end of the experiment, plates were centrifuged at 7000 rpm for 5 min, and spent medium was removed and filter-sterilized with a 0.22 µm filter. Samples were transferred to 1.5 mL microtubes and dried under vacuum using an Eppendorf Vacufuge (Eppendorf, Enfield, CT, USA). Later, they were resuspended in 1 mL LC-MS grade water and 100-fold diluted to be submitted for LC-MS analysis.

## Amygdalin degradation in spent media and cell lysates

*Bifidobacterium* strains wkB204, wkB344, and wkB338 were chosen to investigate the mechanism of amygdalin degradation. They were cultured in 5 mL of MRS broth at 35°C and 5% $CO_2$ overnight. OD was measured for each bacterial culture at 600 nm and diluted to an OD of 0.5 with SDM. Cells were washed twice with SDM. One-hundred µL aliquots of each bacterial suspension were transferred in three biological replicates to 15 mL culture tubes containing 10 mL of SDM with 10 mM amygdalin, 10 mM glucose, or both, or without a carbon source. Samples were incubated at 35°C and 5% $CO_2$ and OD was measured at 600 nm every day for 5 days. At the end of the experiment, amygdalin- and glucose-grown cultures from each strain were centrifuged for 10 min at full speed, and spent medium was separated from the cell pellet.

Spent media of amygdalin- and glucose-grown cultures were filter-sterilized with a 0.22 µm filter, and 2.7 mL aliquots of each sample were transferred in three biological replicates to 15 mL culture tubes containing 0.3 mL of 100 mM amygdalin in SDM to investigate degradation by potential enzymes released into the media.

Bacterial cells of amygdalin- and glucose-grown cultures were washed three times with 1 mL SDM, then the supernatant was removed by centrifugation at full speed for 5 min. Washed cells were lysed with 1 mL of a bacterial protein extraction reagent (B-PER) solution, which consisted of 10 µL of 1 M $MgCl_2$, 20 µL of 0.5 M phenylmethylsulfonyl fluoride (in methanol), and 9970 µL B-PER (Thermo Scientific, catalog number: 78248, lot number: LJ148147A). After 15 min, samples were centrifuged, filter-sterilized with a 0.22 µm filter, and 0.3 mL aliquots of each sample were transferred in three biological replicates to 15 mL culture tubes containing 0.3 mL of 100 mM amygdalin in SDM and 2.4 mL SDM to investigate degradation. Although the cell densities of wkB338 and wkB344 cultures were lower than that of wkB204 (*Figure 3A*), we were still able to collect and concentrate cells through centrifugation for the subsequent experimental steps. Normalization between samples was based on the volume of growth medium, and not on cell mass.

Spent medium and cell lysate of amygdalin-grown cultures only, and 10 mM amygdalin in fresh SDM were used as controls.

All samples were incubated at 35°C and 5% $CO_2$ for 3 days, after which they were 500-fold diluted and submitted for LC-MS analysis.

## Quantification of amygdalin in bacterial cultures

Diluted samples were analyzed using an Agilent 6546 Q-TOF LC-MS with an Agilent Dual Jet Stream electrospray ionization source in negative mode. Chromatographic separations were obtained under gradient conditions by injecting 1 µL onto an Agilent RRHD Eclipse Plus C18 column (50×2.1 mm, 1.8 µm particle size) with an Agilent Zorbax Eclipse Plus C18 narrow bore guard column (12.5×2.1 mm, 5 µm particle size) on an Agilent 1260 Infinity II liquid chromatography system. The mobile phase consisted of eluent A (water + 0.1% formic acid) and eluent B (methanol). The gradient was as follows: held at 5% B from 0 to 1 min, 5% B to 30% B from 1 to 1.5 min, 30% B to 37% B from 1.5 to 9 min, 37% B to 95% B from 9 to 9.1 min, held at 95% B from 9.1 to 12 min, 95% B to 5% B from 12 to 12.1 min, and held at 5% B from 12.1 to 15 min. The flow rate was 0.4 mL/min. The sample tray and column compartment were set to 7°C and 30°C, respectively. The ion source settings were capillary voltage, 3500 V; nozzle voltage, 2000 V; fragmentor voltage, 180 V; drying gas and sheath gas temperature, 350°C; drying gas flow, 10 L/min; sheath gas flow, 11 L/min; nebulizer pressure, 60 lb/in$^2$. Q-TOF data was processed using Agilent MassHunter Qualitative Analysis software. Amygdalin ($C_{20}H_{27}NO_{11}$) and prunasin ($C_{14}H_{17}NO_6$) were observed in the samples with this LC-MS method as $[M-H]^-$ at 456.1511 and 294.0983 Da, as well as $[M+CH_3COO]^-$ at 502.1566 and 340.1038 Da, with a retention time of 2.73 and 3.05 min, respectively. Amygdalin quantification was performed by preparing analytical curves using the area under the amygdalin extracted ion chromatogram peak (20 ppm extraction window) of the following standard solutions prepared from a 1 mM amygdalin stock solution in water: 0.078125, 0.15625, 0.3125, 0.625, 1.25, 2.5, 5, 10, 20, 30, and 40 µM amygdalin. Two analytical curves were prepared: one with the six lower concentrations to calculate amygdalin concentration in 0.1 or 10 mM amygdalin cultures; another curve with the five higher concentrations to calculate amygdalin concentration in 100 mM amygdalin cultures. The linear equations obtained from these analytical curves were used to calculate the concentration of amygdalin in the samples. The concentrations obtained from the linear equation were corrected for the dilution factor. Prunasin quantification was performed similarly, by preparing an analytical curve using the area under the prunasin extracted ion chromatogram peak of the following standard solutions prepared from a 1 mM prunasin stock solution in water: 0.01953125, 0.0390625, 0.078125, 0.15625, 0.3125, 0.625, 1.25, 2.5, 5, and 10 µM prunasin. The linear equation obtained from this analytical curve was used to calculate the concentration of prunasin in the samples. The concentrations obtained from the linear equation were corrected for the dilution factor.

## SDS-PAGE and sample preparation for proteomics analysis

*Bifidobacterium* strain wkB204 was cultured in 5 mL of MRS broth at 35°C and 5% $CO_2$ overnight. OD was measured at 600 nm and adjusted to 0.5 with SDM. Bacterial cells were washed two times with SDM and resuspended in SDM. Two-hundred µL aliquots were transferred in three biological replicates to 250 mL culture flasks containing 100 mL of SDM with 10 mM amygdalin, 10 mM glucose, or without a carbon source. Samples were incubated at 35°C and 5% $CO_2$ and OD was measured at 600 nm every day for 7 days. At the end of the experiment, samples were centrifuged at 7800 rpm for 10 min. Spent media were separated from bacterial cells and concentrated to about 10 mL in under vacuum using an Eppendorf Vacufuge (Eppendorf, Enfield, CT, USA). Then, samples were dialyzed three times in 1 L of exchange buffer (10% glycerol, 1 mM $MgCl_2$, 0.1 M NaCl, 1 mM PMSF, and 25 mM Tris pH 8), after which they were further concentrated with centrifugal concentrators (10 kDa MWCO, Millipore Sigma-Aldrich, Burlington, MA, USA) to a final volume of 1.5 mL. Thirty µL of each concentrated sample were run on a Bolt 4–12% Bis-Tris Plus Gel (Thermo Scientific, catalog number: NW04120BOX, log number: 21022470). Then, concentrated samples from amygdalin- and glucose-grown cultures were submitted for proteomics analysis at the Proteomics facility, UT-Austin. The samples were digested with trypsin, desalted and run on the Dionex LC and Orbitrap Fusion 1 for LC-MS/MS with 1 hr run time and processed by the facility using PD 2.2 and Scaffold proteomics software (Proteome Software, Inc, Portland, OR, USA, version 5.1.2). For protein assignment, we used the amino acid sequences predicted from the wkB204 genome combined with a list of common contaminants for the searches. A basic Scaffold analysis was performed using a custom amino acid sequence database covering the genome of wkB204, a reference database for *Saccharomyces cerevisiae* (because of the yeast extract portion of the SDM used to grow this strain), as well as a list of common contaminants, using min protein: 0.1% false discovery rate.

## Blast search and phylogenetic analysis

A local blast was performed to search for homologous proteins of the GH3 that was detected in wkB204 amygdalin-grown cultures. We used the amino acid sequence of wkB204-GH3 as a query to search for homologous proteins in a custom database containing amino acid sequences of several published bee gut bacterial genomes, including 22 bee gut-associated *Bifidobacterium* strains. We applied a query coverage high-scoring sequence pair percent of 90. wkB204-GH3 and homologous proteins were used to build a phylogenetic tree. Amino acid sequences were aligned using Muscle (*Edgar, 2004*) and used to infer a maximum-likelihood phylogeny (LG model + Gamma4, 100 bootstrap replicates) with PhyML 3.1 (*Guindon et al., 2010*) implemented in SeaView (*Gouy et al., 2010*).

## GH3 gene expression in *Bifidobacterium* strains

One-hundred μL of 0.5 OD cultures of *Bifidobacterium* strains wkB204, wkB344, and wkB338 were transferred in three biological replicates to SDM with 10 mM amygdalin or 10 mM glucose for a final volume of 10 mL. After 5 days, bacterial cultures were centrifuged to separate the supernatant from the cells. Total RNA was extracted from washed bacterial cells using the Quick-RNA Miniprep kit (Zymo Research, Irvine, CA, USA). To that end, bacterial cells were resuspended and lysed in 600 μL of RNA Lysis Buffer, and transferred to a capped vial containing 0.5 mL of 0.1 mm Zirconia beads (BioSpec Products, Bartlesville, OK, USA). Samples were bead-beaten for 2×30 s, centrifuged at 14,000 rpm for 30 s, and transferred to a new 1.5 mL microtube. After this step, extraction followed the protocol provided by Zymo Research. Final RNA samples were dissolved in 50 μL of water and stored at –80°C. RNA concentrations were measured in a Qubit instrument and normalized to 200 ng/μL. Complementary DNA (cDNA) was synthesized using the qScript cDNA Synthesis Kit (QuantaBio, Beverly, MA, USA) following the manufacturer's instructions, and stored at –20°C. cDNA samples were 10-fold diluted to be used as templates for qPCR analyses.

Specific primers targeting a conserved 124 bp region in the GH3 gene found in *Bifidobacterium* strains wkB204 and wkB344 (B-GH3-F: 5'-ctaccgcaatcccgacct-3' and B-GH3-R: 5'-cacctccttgtccact ccc-3') were designed and used to amplify total copies of GH3 gene transcripts in each sample on 384-well plates on a Thermo Fisher QuantStudio 5 instrument. Three technical replicates of 10 μL reactions were carried out for each sample with 5 μL iTaq Universal SYBR Green Supermix (Bio-Rad, Hercules, CA, USA), 0.05 μL (each) 100 μM primer, 3.9 μL $H_2O$, and 1.0 μL template DNA. The cycling conditions consisted of an initial cycle of 50°C for 2 min and 95°C for 2 min, followed by 40 cycles of a two-step PCR of 95°C for 15 s and 60°C for 1 min. Quantification was based on standard curves from amplification of the cloned target sequence in the pGEM-T Easy vector (Promega, Madison, WI, USA). Briefly, genomic DNA of *Bifidobacterium* strain wkB204 was used as a template to amplify the GH3 gene region of interest (124 bp) using the primers B-GH3-F and B-GH3-R. The purified amplicon was ligated into the pGEM-T Easy vector (Promega, Madison, WI, USA). The recombined vector was purified and transformed into *E. coli* strain DH5-alpha competent cells via electroporation using the Gene Pulser Xcell Electroporation System (Bio-Rad, Hercules, CA, USA). The recombined vector was then isolated from an overnight culture using the Monarch Plasmid Miniprep Kit (New England BioLabs, Ipswich, MA, USA), digested by the restriction enzyme ApaI (New England Biolabs, Ipswich, MA, USA), purified, quantified in a Qubit 4 fluorometer (Invitrogen, Waltham, MA, USA) and the final concentration was adjusted so it could be used as a standard for qPCRs.

## Cloning and transformation experiments

*E. coli* strain DH5-alpha was used for gene cloning and *E. coli* strain Rosetta BL21 was used for heterologous expression. LB or MM (*Table 3*) supplemented with 100 μg/mL ampicillin were used for the cultivation. *E. coli* strains were always cultured at 37°C overnight. The vector pET25b (Invitrogen, Waltham, MA, USA) was applied for cloning and expression. First, the vector pET25b-empty was transformed into *E. coli* DH5-alpha cells via electroporation using the Gene Pulser Xcell Electroporation System (Bio-Rad, Hercules, CA, USA). Positive transformants were screened on LB plates with 100 μg/mL ampicillin and by PCR amplification. An overnight culture was used to isolate the vector pET25b-empty (Monarch Plasmid Miniprep Kit, New England BioLabs, Ipswich, MA, USA), which was then dephosphorylated with Antarctic Phosphatase (New England BioLabs, Ipswich, MA, USA) to reduce recyclization. Genomic DNA of *Bifidobacterium* strains wkB204 and wkB344 were used as templates to amplify their respective amygdalin degrading GH3 enzymes by PCR. Specific primers,

GH3-NdeI-F (5'-ttgtttaactttaagaaggagatatacatatggcatcaaggaagttgacagagg-3') and GH3-HindIII-R (5'-agcccgtttgatctcgagtgcggccgcaagcttacccacggtcaccgtca-3') were designed to amplify the whole gene encoding the amygdalin-degrading GH3 enzyme. The PCR products were purified and submitted for Sanger sequencing for confirmation. The purified vector pET25b-empty and the PCR product of wkB204-GH3 (or wkB344-GH3) were digested by the restriction enzymes NdeI and HindIII-HF (both from New England Biolabs, Ipswich, MA, USA) and then ligated to construct the recombinant plasmid pET25b-wkB204-GH3 (or pET25b-wkB344-GH3). The sequence-verified recombinant plasmids were purified and transformed into *E. coli* Rosetta BL21 competent cells via electroporation using the Gene Pulser Xcell Electroporation System (Bio-Rad, Hercules, CA, USA). The empty plasmid was also transformed into *E. coli* Rosetta BL21 competent cells to be used as a control in the experiments. Positive transformants were screened on LB plates with 100 µg/mL ampicillin and by PCR amplification, and bacterial stocks were made from single cell, overnight cultures.

### GH3 gene expression in transformed *E. coli* strains

One-hundred µL of 0.5 OD cultures of *E. coli* Rosetta BL21 cells carrying pET25b-empty, pET25b-wkB204-GH3, or pET25b-wkB344-GH3 were transferred in three biological replicates to 5 mL LB broth supplemented with 100 µg/mL ampicillin and 100 µg/mL isopropyl β-D-1-thiogalactopyranoside (IPTG). Bacterial cultures were grown overnight at 37°C, after which cells were separated from the supernatant by centrifugation. Total RNA was extracted from washed cells using the Quick-RNA Miniprep kit (Zymo Research, Irvine, CA, USA), cDNA was synthesized using the qScript cDNA Synthesis Kit (QuantaBio, Beverly, MA, USA), and qPCR was performed using the primers B-GH3-F and B-GH3-R and following the protocol described in the 'GH3 gene expression in *Bifidobacterium* strains' section.

### Amygdalin and prunasin degradation in cell lysates of transformed *E. coli*

In vitro experiments were performed with transformed *E. coli* Rosetta BL21 cells carrying pET25b-empty, pET25b-wkB204-GH3, or pET25b-wkB344-GH3. To that end, transformants were grown overnight at 37°C in LB supplemented with 100 µg/mL ampicillin and 100 µg/mL IPTG. OD was adjusted to 1 and cells were washed twice with MM (*Table 2*). Five mL of 1 OD washed bacterial cultures were transferred to 5 mL Falcon tubes, centrifuged for 10 min at 7000 rpm, supernatant was removed, and cells were resuspended in 5 mL MM. Bacterial cells were centrifuged again and media was removed. Washed cells were lysed with 1 mL of B-PER solution, as described above, for 15 min at room temperature, after which 4 mL of MM was added. Samples were filter-sterilized with a 0.22 µm filter and dialyzed in centrifugal concentrators (10 kDa MWCO, Millipore Sigma-Aldrich, Burlington, MA, USA) for 20 min. After dialysis, the final volume of concentrated samples was adjusted to 5 mL with MM. 0.5 mL aliquots of each sample were transferred in three biological replicates to 1.5 mL tubes containing 0.5 mL of 0.2 mM amygdalin or 0.2 mM prunasin in MM to investigate degradation. 0.1 mM amygdalin in fresh MM or 0.1 mM prunasin in fresh MM were used as controls. Samples were incubated at 37°C for 5 days, after which they were 10-fold diluted and submitted for LC-MS analysis.

### In vivo experiment to investigate amygdalin degradation in the bee gut

Late-stage pupae (with eyes pigmented but lacking movement) of *A. mellifera* female workers were aseptically removed from a brood frame from a hive kept at UT-Austin. Pupae were placed on Kimwipes in sterile plastic bins and placed in an incubator at 35°C and ~60% relative humidity to simulate hive conditions until emerging as adults. After 3 days, newly emerged workers (NEWs), which lack their normal microbiota, were transferred to cup cages containing sterile sucrose syrup and sterile bee bread. Approximately 400 NEWs were randomly divided into four groups which were fed sterile sucrose syrup and specific treatments as described below. Group 1 was exposed to sterile pollen, and therefore the bees remained as MD. Group 2 was exposed to a fresh bee gut homogenate mixed with sterile pollen, and therefore the bees acquired the normal microbiota. The gut homogenate was prepared by aseptically pulling out the guts from 10 healthy workers from the same hive and mixing with equal proportions of 1× PBS and sterile sucrose syrup (5 mL total volume), and 200 µL of gut homogenate were transferred to sterile pollen and provided to the bees in each cup cage. Groups 3 and 4 were exposed to a *Bifidobacterium* wkB204 or wkB344 bacterial suspension, respectively. Each

bacterial strain was cultured in SDM at 35°C and 5% $CO_2$ overnight. The 600 nm OD of each bacterial culture was measured, cells were washed with 1× PBS, and diluted to a concentration of 0.5 OD in equal proportions of 1× PBS and sterile sucrose syrup. Two-hundred µL of bacterial suspension were transferred to the bee bread provided to the bees in each cup cage. After 5 days, which is sufficient time for establishment of the gut microbiota (*Powell et al., 2014*), bees were transferred to 0.5 mL vials with tips cut off, then starved for 6 hr, after which they were hand-fed with 5 µL of 1 mM amygdalin in sterile sugar syrup. They were kept in the same vial for 18 hr, after which they were frozen until further processing. The control group consisted of unexposed bees that were mixed with 5 µL of 1 mM amygdalin at the beginning of sample processing to determine the amount of amygdalin that would be detected before any degradation event could occur. Three bees from each group were thawed and aseptically dissected to obtain the following bee body compartments: midgut, hindgut, and bee body carcass. These samples were homogenized with 1 mL LC-MS grade water and submitted for LC-MS analyses.

## In vivo experiment to investigate the effects of amygdalin on honey bees and their gut microbiota

First experiment. A brood frame was collected from a honey bee hive at UT-Austin, transferred to a frame cage and placed in an incubator at 35°C and ~60% relative humidity to simulate hive conditions until adults emerged. One -day-old bees were randomly divided into four groups, each being treated with sucrose syrup, 0.01 mM amygdalin dissolved in sucrose syrup, 0.1 mM amygdalin dissolved in sucrose syrup, or 1 mM amygdalin dissolved in sucrose syrup. A gut homogenate (200 µL) was added to the bee bread provided to each cup cage, enabling colonization by the full gut microbiota, as in *Motta and Moran, 2020b*. Fifteen bees were sampled from each group after 1 week of treatment and stored at –80°C. Each group consisted of 4 cup cages each containing 40 bees. Survival rates were monitored and dead bees were removed in a daily census.

Second experiment. Two other brood frames were collected from a different hive at UT-Austin, and bees were allowed to emerge under lab conditions similar to the first experiment. One -day-old bees were randomly split into 36 cup cages, with 29–36 bees each, and divided into two groups to be treated (microbiota-colonized) or not (MD) with a gut homogenate solution. Each main group was divided into three subgroups and fed sucrose (control), 0.01 mM amygdalin in sucrose syrup, or 0.1 mM amygdalin in sucrose syrup. Sucrose syrup was filter-sterilized. Survival rates were monitored, and dead bees were removed in a daily census.

## DNA extraction, qPCR analysis, and 16S rRNA amplicon sequencing

Sampled honey bees from the first experiment were placed in sterile Falcon tubes and transferred to a freezer at –80°C. DNA was extracted from individual guts, following a previously described protocol (*Kwong et al., 2017b*). Final DNA samples were 10-fold diluted to be used as templates for qPCR analyses, as described in *Motta et al., 2018*, and for 16S rRNA library preparation and sequencing, as described in *Motta et al., 2020a*.

## Acknowledgements

Thanks to the members of the Nancy Moran and Howard Ochman labs, especially Kim Hammond for maintaining hives, Eli Powell for lab assistance, and former member Margaret Steele for providing *Gilliamella* strains M1-2G and M6-3G. Thanks to Brianna Flynn for helping isolate *Lactobacillus* strains HB-1, HB-2, HB-C2, HB-D10, and Ryan Arnott for preparing 16S rRNA gene PCR amplicons from these strains for Sanger sequencing analyses.

Thanks to Maria Person, Michelle Gadush, and Peter Faull at the UT Austin Center for Biomedical Research Support Biological Mass Spectrometry Facility (RRID:SCR_021728) for processing protein samples and providing feedback on the raw data.

This work was supported by the USDA National Institute of Food and Agriculture (grant number 2018-67013-27540).

## Additional information

### Funding

| Funder | Grant reference number | Author |
|--------|------------------------|--------|
| National Institute of Food and Agriculture | 2018-67013-27540 | Nancy Moran |

The funders had no role in study design, data collection and interpretation, or the decision to submit the work for publication.

### Author contributions

Erick VS Motta, Conceptualization, Data curation, Formal analysis, Validation, Investigation, Visualization, Methodology, Writing - original draft, Project administration, Writing - review and editing; Alejandra Gage, Thomas E Smith, Kristin J Blake, Investigation, Methodology, Writing - review and editing; Waldan K Kwong, Ian M Riddington, Investigation, Writing - review and editing; Nancy Moran, Resources, Supervision, Funding acquisition, Writing - review and editing

### Author ORCIDs

Erick VS Motta ⓘ http://orcid.org/0000-0001-9360-4353
Nancy Moran ⓘ http://orcid.org/0000-0003-2983-9769

### Decision letter and Author response

Decision letter https://doi.org/10.7554/eLife.82595.sa1
Author response https://doi.org/10.7554/eLife.82595.sa2

## Additional files

### Supplementary files

• MDAR checklist

### Data availability

Bacterial strains are available by request from the Moran Lab. The complete genome sequence of strain BI-2.5 has been deposited at DDBJ/ENA/GenBank under the accession CP031513. The genome assemblies for strains BI-1.1, LV-8.1, BI-4G, L5-31, OCC3 and wkB204 have been deposited at DDBJ/ENA/GenBank under the accessions QOCR00000000, QOCS00000000, QOCU00000000, QOCT00000000, QOCV00000000 and JAFMNU020000000, respectively. 16S rRNA amplicon sequencing data are available at NCBI BioProject PRJNA865802.

The following datasets were generated:

| Author(s) | Year | Dataset title | Dataset URL | Database and Identifier |
|-----------|------|---------------|-------------|-------------------------|
| Motta EVS, Kwong WK, Moran NA | 2022 | Bifidobacterium asteroides strain wkB204, whole genome shotgun sequencing project | https://www.ncbi.nlm.nih.gov/nuccore/JAFMNU000000000.2 | NCBI Nucleotide, JAFMNU000000000.2 |
| Motta EVS, Gage A, Smith TE, Blake KJ, Kwong WK, Riddington IM, Moran NA | 2022 | Cooperative host-microbe metabolism of a plant toxin in bees | https://www.ncbi.nlm.nih.gov/bioproject/PRJNA865802/ | NCBI BioProject, PRJNA865802 |

The following previously published datasets were used:

| Author(s) | Year | Dataset title | Dataset URL | Database and Identifier |
|---|---|---|---|---|
| Motta EVS, Moran NA | 2018 | Lactobacillus bombicola strain OCC3, whole genome shotgun sequencing project | https://www.ncbi.nlm.nih.gov/nuccore/QOCV00000000 | NCBI Nucleotide, QOCV00000000 |
| Motta EVS, Moran NA | 2018 | Lactobacillus bombicola strain BI-4G, whole genome shotgun sequencing project | https://www.ncbi.nlm.nih.gov/nuccore/QOCU00000000 | NCBI Nucleotide, QOCU00000000 |
| Motta EVS, Moran NA | 2018 | Bombilactobacillus bombi strain LV-8.1, whole genome shotgun sequencing project | https://www.ncbi.nlm.nih.gov/nuccore/QOCS00000000 | NCBI Nucleotide, QOCS00000000 |
| Motta EVS, Moran NA | 2018 | Bombilactobacillus bombi strain BI-1.1, whole genome shotgun sequencing project | https://www.ncbi.nlm.nih.gov/nuccore/QOCR00000000 | NCBI Nucleotide, QOCR00000000 |
| Motta EVS, Kwong WK, Moran NA | 2018 | Bombilactobacillus bombi strain BI-2.5 chromosome, complete genome | https://www.ncbi.nlm.nih.gov/nuccore/CP031513.1/ | NCBI Nucleotide, CP031513 |

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
