## [Editor Report]

The manuscript makes an important contribution to understanding the roles of the bee host and microbiome in degrading amygdalin, a dietary secondary metabolite. Several bacterial strains and their enzymes responsible for the deglycosylation of amygdalin are identified. Conclusions are reached convincingly through a comprehensive combination of in vitro and in vivo experiments including gene-expression analysis, proteomics, HPLC-MS, and the use of recombinant *E. coli* to test enzyme function. The consequences of microbial-derived amygdalin metabolisation on host health remain uncertain from the experiments conducted, but this work should stimulate future research into the importance of secondary metabolite processing by the microbiome on insect host health.

---

## [Decision Letter]

**Decision letter after peer review:**

Thank you for submitting your article "Cooperative host-microbe metabolism of a plant toxin in bees" for consideration by *eLife*. Your article has been reviewed by 3 peer reviewers, including Hauke Koch as Reviewing Editor and Reviewer #1, and the evaluation has been overseen by Wendy Garrett as the Senior Editor.

Essential revisions:

All three referees thought the manuscript was important with convincing evidence to support conclusions. Requests for revisions included:

1) More careful wording in parts of the manuscript around the effects of the observed microbiome-derived degradation of amygdalin and emphasis on what future experiments would need to be done to determine effects on host health. The use of the word "cooperation" and similar language in the title and parts of the discussion should be revised.

2) Revisions of the discussion to remove redundancy with the Results section and include context and parallels to non-bee work.

*Reviewer #2 (Recommendations for the authors):*

While the possible consequences for host fitness are indeed interesting, the authors do a good job pointing to future directions building on current findings. Most interestingly, potentially toxic breakdown products such as hydrogen cyanide are proposed to confer protection against pathogen infection. This, I think, is where more context is necessary, and for which potential parallels to non-bee work can be highlighted (See de Roode and Hunter, 2019). Finally, a few paragraphs in the discussion are redundant with the Results section (e.g. lines 387-396) and can be streamlined without affecting the clarity. This manuscript was a pleasure to read, adding a nice case to our understanding of symbiont-mediated degradation of plant toxins.

*Reviewer #3 (Recommendations for the authors):*

I enjoyed reading this manuscript, which does a good job of demonstrating and dissecting the strain-specific metabolism of amygdalin.

I find the discussion of potential adaptive value when it comes to parasite resistance overly speculative, especially as there is no demonstration of clear negative or beneficial health effects of the different breakdown products on the bees and only a limited number of strains exhibit the ability the metabolize amygdalin and its metabolites. Could this not be a product of chance or another process?

I think a clear demonstration of the effects in microbiota complete and free bees of the different breakdown products, for example using a dose-response curve approach for survival, would add considerably to the interpretation of the results.

---

## [Author Response]

Essential revisions:All three referees thought the manuscript was important with convincing evidence to support conclusions. Requests for revisions included:1) More careful wording in parts of the manuscript around the effects of the observed microbiome-derived degradation of amygdalin and emphasis on what future experiments would need to be done to determine effects on host health. The use of the word "cooperation" and similar language in the title and parts of the discussion should be revised.

We have clarified the implications for impacts of the microbial degradation on bee health, avoiding unwarranted conclusions about benefits for bees (lines 326-333, 399-402, additional Figure 9 —figure supplement 1).

We have included two additional paragraphs (lines 474-497) to place the study in the context of other studies on bee-microbiota interactions and to discuss potential future experiments to investigate the consequences of amygdalin degradation on bee health.

We agree that “cooperative” does not fit here, and we have removed the word “cooperative” from the title and replaced it with “jointly” in one instance in the discussion (line 485).

2) Revisions of the discussion to remove redundancy with the Results section and include context and parallels to non-bee work.

We have reorganized the discussion to remove redundancy with the Results section and have included additional context relating the work to other host-microbe systems, as suggested by the reviewers (lines 336-349, 432-443, 451-4561).

Reviewer #2 (Recommendations for the authors):While the possible consequences for host fitness are indeed interesting, the authors do a good job pointing to future directions building on current findings. Most interestingly, potentially toxic breakdown products such as hydrogen cyanide are proposed to confer protection against pathogen infection. This, I think, is where more context is necessary, and for which potential parallels to non-bee work can be highlighted (See de Roode and Hunter, 2019).

We agree with this comment and have added some discussion regarding the protection that secondary metabolites can confer to other insects (lines 432-443).

Finally, a few paragraphs in the discussion are redundant with the Results section (e.g. lines 387-396) and can be streamlined without affecting the clarity.

We agree and have rearranged the discussion to avoid repetition of the results.

This manuscript was a pleasure to read, adding a nice case to our understanding of symbiont-mediated degradation of plant toxins.

Thank you!

Reviewer #3 (Recommendations for the authors):I enjoyed reading this manuscript, which does a good job of demonstrating and dissecting the strain-specific metabolism of amygdalin.

Thank you!

I find the discussion of potential adaptive value when it comes to parasite resistance overly speculative, especially as there is no demonstration of clear negative or beneficial health effects of the different breakdown products on the bees and only a limited number of strains exhibit the ability the metabolize amygdalin and its metabolites. Could this not be a product of chance or another process?

We don’t believe amygdalin degradation occurs by chance in the bee gut, but we agree that it may occur simply for the benefit of the microbes themselves, for example, to use it as an additional carbon substrate. We have altered the wording in numerous places to avoid implication of adaptive value to bees.

Although we only detected a limited number of strains that can degrade amygdalin in our in vitro experiments, this is mainly because we only tested a limited number of strains available in the laboratory. Moreover, we were only able to grow a subset of strains in semi-defined media, and strains of *Gilliamella* and *Lactobacillus* nr. *melliventris* had to be cultivated in rich media, which may have masked their potential to degrade amygdalin. We note that the in vivo experiment demonstrates that the native gut microbiota can break down both amygdalin and prunasin, which was not degraded in microbiota-deprived bees. This suggests that the native strain diversity in many or all bees does include ability to degrade amygdalin (CV bees versus MD bees in Figure 8).

I think a clear demonstration of the effects in microbiota complete and free bees of the different breakdown products, for example using a dose-response curve approach for survival, would add considerably to the interpretation of the results.

We agree that further work should be done to fully understand the consequences of amygdalin degradation for bees and have provided some context in lines 484-497. We also now include additional results that show that field-relevant doses of amygdalin do not affect survival rates of microbiota-deprived or microbiota-colonized bees (Figure 9 —figure supplement 1).